# Dynamic localization of the chromosomal passenger complex in trypanosomes is controlled by the orphan kinesins KIN-A and KIN-B

**Daniel Ballmer[1,2], Bungo Akiyoshi[1,2]\***

[1]Department of Biochemistry, University of Oxford, Oxford, United Kingdom; [2]The Wellcome Centre for Cell Biology, Institute of Cell Biology, School of Biological Sciences, Edinburgh, United Kingdom

**\*For correspondence:**
bungo.akiyoshi@ed.ac.uk

**Competing interest:** The authors declare that no competing interests exist.

**Abstract** The chromosomal passenger complex (CPC) is an important regulator of cell division, which shows dynamic subcellular localization throughout mitosis, including kinetochores and the spindle midzone. In traditional model eukaryotes such as yeasts and humans, the CPC consists of the catalytic subunit Aurora B kinase, its activator INCENP, and the localization module proteins Borealin and Survivin. Intriguingly, Aurora B and INCENP as well as their localization pattern are conserved in kinetoplastids, an evolutionarily divergent group of eukaryotes that possess unique kinetochore proteins and lack homologs of Borealin or Survivin. It is not understood how the kinetoplastid CPC assembles nor how it is targeted to its subcellular destinations during the cell cycle. Here, we identify two orphan kinesins, KIN-A and KIN-B, as bona fide CPC proteins in *Trypanosoma brucei*, the kinetoplastid parasite that causes African sleeping sickness. KIN-A and KIN-B form a scaffold for the assembly of the remaining CPC subunits. We show that the C-terminal unstructured tail of KIN-A interacts with the KKT8 complex at kinetochores, while its N-terminal motor domain promotes CPC translocation to spindle microtubules. Thus, the KIN-A:KIN-B complex constitutes a unique 'two-in-one' CPC localization module, which directs the CPC to kinetochores from S phase until metaphase and to the central spindle in anaphase. Our findings highlight the evolutionary diversity of CPC proteins and raise the possibility that kinesins may have served as the original transport vehicles for Aurora kinases in early eukaryotes.

## eLife assessment

This **important** study identifies the mitotic localization mechanism for Aurora B and INCENP (parts of the chromosomal passenger complex, CPC) in *Trypanosoma brucei*. The mechanism differs from that in the more commonly studied opisthokonts and is supported by **compelling** RNAi and imaging experiments, targeted mutations, immunoprecipitations with crosslinking/mass spec, and AlphaFold interaction predictions. The findings will be of interest to cell biologists working on cell division, parasitologists, and those interested in the evolution of mitotic mechanisms.

## Introduction

During cell division, duplicated genetic material must be distributed equally into two daughter cells. The Aurora B kinase is a key mitotic regulator widely conserved among eukaryotes (*Hochegger et al., 2013*). It undergoes dynamic localization changes throughout mitosis to enable the spatially restricted phosphorylation of substrates involved in chromosome alignment, chromosome bi-orientation, spindle

assembly checkpoint (SAC) signaling, and cytokinesis (*Carmena et al., 2012*). In early mitosis, Aurora B is first detected on chromosome arms and during prometaphase becomes enriched at centromeres, where it destabilizes incorrect kinetochore-microtubule attachments (*Krenn and Musacchio, 2015*). Upon anaphase onset, Aurora B translocates to the spindle midzone, and during cytokinesis associates with the equatorial cortex to regulate cell abscission (*Adams et al., 2000*; *Cooke et al., 1987*; *Trivedi and Stukenberg, 2016*).

The dynamic localization pattern of the Aurora B kinase is in part achieved through its association with a scaffold comprised of inner centromere protein (INCENP), Borealin, and Survivin (*Adams et al., 2001*; *Adams et al., 2000*; *Gassmann et al., 2004*; *Romano et al., 2003*; *Sampath et al., 2004*; *Vader et al., 2006*; *Wheatley et al., 2001*). Together, these proteins form a tetrameric complex referred to as the chromosomal passenger complex (CPC). The CPC can be partitioned into two functional modules: The 'catalytic module' and the 'localization module'. The catalytic module is composed of Aurora B in complex with the IN-box at the INCENP C-terminus, which is required for full activation of the Aurora B kinase (*Bishop and Schumacher, 2002*). The localization module comprises Borealin, Survivin, and the N-terminus of INCENP, which are connected to one another via a three-helical bundle (*Jeyaprakash et al., 2011*; *Jeyaprakash et al., 2007*; *Klein et al., 2006*). The two modules are linked by the central region of INCENP, composed of an intrinsically disordered domain and a single alpha helical (SAH) domain. INCENP harbors microtubule-binding domains within the N-terminus and the central SAH domain, which play key roles for CPC localization and function (*Cormier et al., 2013*; *Fink et al., 2017*; *Kang et al., 2001*; *Mackay et al., 1993*; *Nakajima et al., 2011*; *Noujaim et al., 2014*; *Samejima et al., 2015*; *van der Horst et al., 2015*; *Wheatley et al., 2001*; *Wheelock et al., 2017*).

In vertebrates, recruitment of the CPC to centromeric chromatin depends on two pathways, involving the Haspin and Bub1 kinases. Haspin phosphorylates histone H3 on Thr3 (H3T3ph), which is recognized by the baculovirus IAP repeat (BIR) domain of Survivin (*Kelly et al., 2010*; *Wang et al., 2010*; *Yamagishi et al., 2010*). H3T3ph is initially found along the entire length of chromosomes between sister chromatids but becomes enriched at the inner centromere (the space between sister kinetochores) during late prophase. In contrast, the kinetochore-associated Bub1 kinase phosphorylates histone H2A on Thr120 (H2AT120ph) (*Kawashima et al., 2010*). H2AT120ph recruits Shugoshin-like proteins (Sgo1 and Sgo2), which in turn are bound by Borealin (*Tsukahara et al., 2010*; *Yamagishi et al., 2010*). Recently, Sgo1 has also been demonstrated to interact with the BIR domain of Survivin through an N-terminal histone H3-like motif (*Abad et al., 2022*; *Jeyaprakash et al., 2011*). The interactions of Borealin and Survivin with Sgo1 form the basis for a kinetochore-proximal pool of the CPC which is distinct from the inner centromere pool (*Broad et al., 2020*; *Hadders et al., 2020*; *Liang et al., 2020*).

In most studied eukaryotes, ranging from yeast to humans, kinetochore assembly is scaffolded by a centromere-specific histone H3 variant, CENP-A (*Allshire and Karpen, 2008*; *Black and Cleveland, 2011*; *Hori and Fukagawa, 2012*; *Maddox et al., 2012*; *Westhorpe and Straight, 2013*). An assembly of inner kinetochore protein complexes, referred to as the constitutive centromere-associated network, interacts with centromeric CENP-A chromatin throughout the cell cycle and provides a platform for recruitment of the outer kinetochore KNL1/Mis12 complex/Ndc80 complex network that has microtubule-binding activity (*Cheeseman et al., 2006*; *Foltz et al., 2006*; *Izuta et al., 2006*; *Okada et al., 2006*). Some of these kinetochore proteins are present in nearly all sequenced eukaryotes, suggesting that key principles of chromosome segregation are widely shared among eukaryotes (*Drinnenberg and Akiyoshi, 2017*; *van Hooff et al., 2017*; *Meraldi et al., 2006*; *Tromer et al., 2019*). However, a unique set of kinetochore proteins (KKT1–20, KKT22–25, KKIP1–12) are present in kinetoplastids (*Brusini et al., 2021*; *Akiyoshi and Gull, 2014*; *D'Archivio and Wickstead, 2017*; *Nerusheva et al., 2019*; *Nerusheva and Akiyoshi, 2016*), a group of flagellated protists that are highly divergent from commonly studied eukaryotes (*Cavalier-Smith, 2010*). *Trypanosoma brucei*, *Trypanosoma cruzi*, and *Leishmania* spp. are causative agents of African trypanosomiasis, Chagas disease, and leishmaniasis, respectively, and as such pose a serious threat to public health and prosperity across the tropics and subtropics (*Stuart et al., 2008*; *WHO, 2017*).

Despite the absence of canonical kinetochore components (*Berriman et al., 2005*; *Lowell and Cross, 2004*), Aurora kinases are conserved in kinetoplastids. Early studies suggested that the Aurora B homolog (Aurora B[AUK1]) in *T. brucei* forms a complex with chromosomal passenger complex proteins

1 and 2 (CPC1 and CPC2) and plays a crucial role in mitosis and cytokinesis (*Li et al., 2008*; *Tu et al., 2006*). CPC1 was later found to be a divergent INCENP homolog (hereafter referred to as INCENP$^{CPC1}$) based on the presence of a conserved C-terminal IN-box (*Hu et al., 2014*). However, INCENP$^{CPC1}$ lacks the central SAH domain and N-terminal residues, which in other eukaryotes interact with Survivin and Borealin. In addition, two orphan kinesins, KIN-A and KIN-B, have been proposed to transiently associate with Aurora B$^{AUK1}$ during mitosis (*Li, 2012*; *Li et al., 2008*). Although homologs of the 'localization module' proteins Survivin and Borealin have not been identified in kinetoplastids (*Komaki et al., 2022*), the trypanosome CPC displays a dynamic localization pattern similar to that of the metazoan CPC (*Li et al., 2008*): Aurora B$^{AUK1}$, INCENP$^{CPC1}$, and CPC2 localize to kinetochores in early mitosis and then translocate to the central spindle upon anaphase onset. From late anaphase onward, an additional population of CPC proteins is detectable at the tip of the new flagellum attachment zone (FAZ), the point of cytokinesis initiation in *T. brucei*. It is presently not understood how the CPC assembles in these evolutionarily divergent eukaryotes nor how its localization dynamics are regulated during the cell cycle.

Here, by combining biochemical, structural, and cell biological approaches in procyclic form *T. brucei*, we show that the trypanosome CPC is a pentameric complex comprising Aurora B$^{AUK1}$, INCENP$^{CPC1}$, CPC2, and the two orphan kinesins KIN-A and KIN-B. KIN-A and KIN-B interact via their coiled-coil domains to form a subcomplex within the CPC, which serves as a scaffold for the catalytic module (Aurora B$^{AUK1}$ + INCENP$^{CPC1}$). The C-terminal unstructured tail of KIN-A directs kinetochore localization of the CPC from S phase to metaphase, while the N-terminal motor domain promotes the central spindle enrichment in anaphase. Furthermore, we identify the KKT7-KKT8 complex pathway as the main kinetochore recruitment arm of the trypanosome CPC.

## Results

### KIN-A and KIN-B are bona fide CPC proteins in trypanosomes

To identify additional interactors of the CPC in trypanosomes, we performed immunoprecipitation followed by liquid chromatography tandem mass spectrometry (IP-MS) of endogenously YFP-tagged Aurora B$^{AUK1}$ (*Figure 1—figure supplement 1A*, *Supplementary files 1 and 2*). Besides Aurora B$^{AUK1}$, INCENP$^{CPC1}$, and CPC2, we observed notable enrichment of two orphan kinesins, KIN-A and KIN-B (*Wickstead and Gull, 2006*), as reported previously (*Li et al., 2008*). Both KIN-A and KIN-B were also highly enriched in immunoprecipitates of ectopically expressed GFP-INCENP$^{CPC1}$. Vice versa, IP-MS of GFP-tagged KIN-A and KIN-B identified Aurora B$^{AUK1}$, INCENP$^{CPC1}$, and CPC2 as top hits (*Figure 1A* and *Supplementary file 2*).

We next assessed the localization dynamics of fluorescently tagged KIN-A and KIN-B over the course of the cell cycle (*Figure 1B–E*). *T. brucei* possesses two DNA-containing organelles, the nucleus ('N') and the kinetoplast ('K'). The kinetoplast is an organelle found uniquely in kinetoplastids, which contains the mitochondrial DNA and replicates and segregates prior to nuclear division. The 'KN' configuration serves as a good cell cycle marker (*Siegel et al., 2008*; *Woodward and Gull, 1990*). To our surprise, KIN-A-YFP and GFP-KIN-B exhibited a CPC-like localization pattern similar to that of Aurora B$^{AUK1}$: Both kinesins localized to kinetochores from S phase to metaphase, and then translocated to the central spindle in anaphase (*Figure 1C–E*). Moreover, like Aurora B$^{AUK1}$, a population of KIN-A and KIN-B localized at the new FAZ tip from late anaphase onward (*Figure 1—figure supplement 1B and C*). This was unexpected, because KIN-A and KIN-B were previously reported to localize to the spindle but not to kinetochores or the new FAZ tip (*Li et al., 2008*). These data suggest that KIN-A and KIN-B are bona fide CPC proteins in trypanosomes, associating with Aurora B$^{AUK1}$, INCENP$^{CPC1}$, and CPC2 throughout the cell cycle.

A bioinformatic search for homologs of CPC proteins within Euglenozoa revealed that both KIN-A and KIN-B are present in trypanosomatids and bodonids, with KIN-A homologs detectable even in prokinetoplastids (*Figure 1F*) (Materials and methods). CPC2, on the other hand, was detectable only within trypanosomatids. Aurora B$^{AUK1}$ and INCENP$^{CPC1}$ are present in kinetoplastids as well as in diplonemids and euglenids (sister groups of kinetoplastids). Interestingly, homologs of Borealin, Survivin, KIN-A, or KIN-B were not detectable in diplonemids or euglenids, raising a possibility that these organisms may also possess 'non-canonical' CPC proteins. We conclude that the KIN-A and

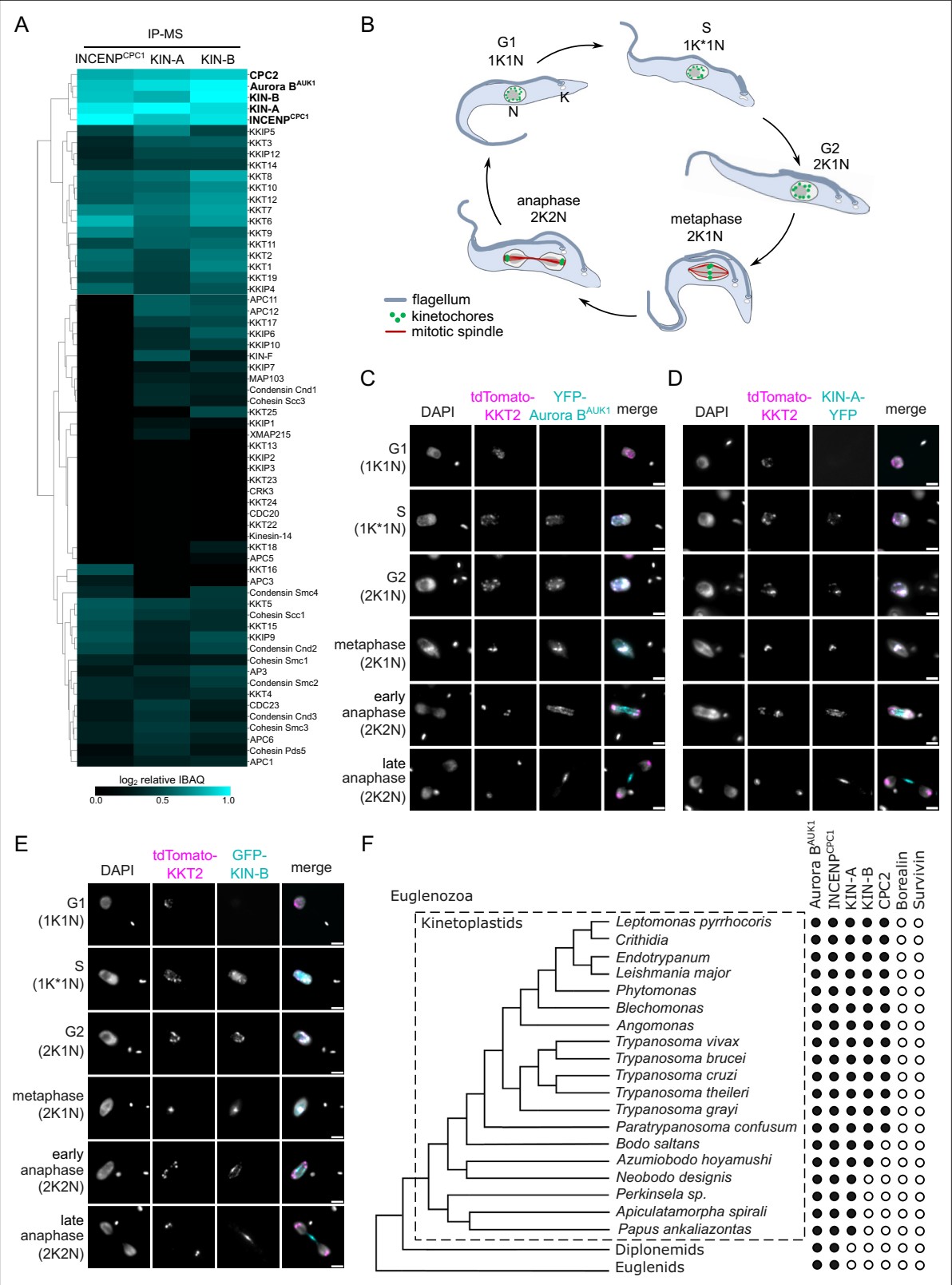

**Figure 1.** KIN-A and KIN-B are bona fide chromosomal passenger complex (CPC) proteins in *T. brucei*. (**A**) Clustered heatmap showing enrichment (log2 intensity-based absolute quantification [IBAQ]) of mitotic proteins co-purifying with ectopically expressed GFP-INCENP^CPC1, GFP-KIN-A, and GFP-KIN-B. The heatmap was generated using the Python Seaborn library using WPGMA clustering. Cell lines: BAP2190, BAP2286, BAP2288. Immunoprecipitation was performed using anti-GFP antibodies. See *Supplementary file 2* for all proteins identified by mass spectrometry. (**B**) Cartoon depicting the

Figure 1 continued

kinetoplast (**K**)/nucleus (**N**) configuration throughout the cell cycle in procyclic *T. brucei*, with K* denoting an elongated kinetoplast. The kinetoplast is an organelle found uniquely in kinetoplastids, which contains the mitochondrial DNA and replicates and segregates prior to nuclear division. The KN configuration serves as a cell cycle marker (*Siegel et al., 2008*; *Woodward and Gull, 1990*). (**C–E**) Representative fluorescence micrographs showing the dynamic localization of YFP-Aurora B[AUK1] (**C**), KIN-A-YFP (**D**), and GFP-KIN-B (**E**) over the course of the cell cycle. Kinetochores are marked with tdTomato-KKT2. DNA was stained with DAPI. Cell lines: BAP1515, BAP3066, BAP2288. Scale bars, 2 µm. (**F**) Phylogenetic tree of kinetoplastids, diplonemids, and euglenids along with the presence (black dots)/absence (white dots) patterns of CPC components. The phylogenetic tree of Euglenozoa is based on *Butenko et al., 2020*.

The online version of this article includes the following figure supplement(s) for figure 1:

**Figure supplement 1.** KIN-A and KIN-B are bona fide chromosomal passenger complex (CPC) proteins in *T. brucei*.

KIN-B kinesins are highly conserved within kinetoplastids, constituting integral components of the CPC in this evolutionary divergent group of eukaryotes.

## KIN-A and KIN-B promote kinetochore localization of the CPC

To investigate which subunits of the CPC are responsible for its kinetochore targeting, we performed a series of RNAi experiments (*Figure 2A–E*). Because CPC2 is poorly conserved among kinetoplastids (*Figure 1F*) and depletion of CPC2 using two different hairpin RNAi constructs (*Supplementary file 1*) was inefficient, we did not include CPC2 in these experiments. As previously reported (*Li et al., 2008*), depletion of Aurora B[AUK1], INCENP[CPC1], KIN-A, or KIN-B resulted in a prominent growth defect (*Figure 2—figure supplement 1A*) with cells arresting in G2/M (2K1N) (*Figure 2—figure supplement 1B*). Knockdown of Aurora B[AUK1] did not affect the kinetochore localization of YFP-tagged INCEN-P[CPC1], KIN-A, or KIN-B (*Figure 2A and C–E*). Knockdown of INCENP[CPC1] caused delocalization of Aurora B[AUK1] but not of KIN-A or KIN-B (*Figure 2A, B, D, and E*). In contrast, both Aurora B[AUK1] and INCENP[CPC1] were delocalized upon depletion of KIN-A or KIN-B (*Figure 2A–C*). RNAi against KIN-A disrupted KIN-B localization and vice versa (*Figure 2A, D, and E*). Moreover, total protein levels of KIN-B were affected by depletion of KIN-A (*Figure 2—figure supplement 1C*), suggesting that the interaction with KIN-A is required to stabilize KIN-B.

We next ectopically expressed GFP-tagged truncations of KIN-A and KIN-B to assess which domains promote their kinetochore targeting. Both KIN-A and KIN-B contain an N-terminal kinesin motor domain followed by several predicted coiled-coil motifs (*Li et al., 2008*), although KIN-B is predicted to be an inactive motor (*Wickstead and Gull, 2006*). In addition, KIN-A has a long, disordered C-terminal tail (see below). Unlike full-length KIN-B, KIN-B[2-316] (inactive motor domain) failed to enrich at kinetochores and was instead found in the nucleolus (*Figure 2—figure supplement 1D and E*). KIN-A[2-309] (motor domain) was also primarily detected in the nucleolus, although we observed additional spindle and weak kinetochore-like signal in some metaphase cells (*Figure 2G*). By contrast, both KIN-A[310-862] (coiled-coil domain+C-terminal disordered tail) and KIN-B[317-624] (coiled-coil domain) clearly localized to kinetochores from S phase to metaphase (*Figure 2H*; *Figure 2—figure supplement 1F*). Intriguingly, unlike endogenously YFP-tagged KIN-A, ectopically expressed GFP fusions of both full-length KIN-A and KIN-A[310-862] localized at kinetochores even in anaphase (*Figure 2F and H*). Weak anaphase kinetochore signal was also detectable for KIN-B[317-624] (*Figure 2—figure supplement 1F*). GFP fusions of the central coiled-coil domain or the C-terminal disordered tail of KIN-A did not localize to kinetochores (data not shown). These results show that kinetochore localization of the CPC is mediated by KIN-A and KIN-B and requires both the central coiled-coil domain and the C-terminal disordered tail of KIN-A.

## Structural model of the trypanosome CPC

To gain structural insights into the trypanosome CPC, we used AlphaFold2 (AF2) (*Jumper et al., 2021*; *Mirdita et al., 2022*) to predict the overall structure of the trypanosome CPC by testing combinations of full-length Aurora B[AUK1], INCENP[CPC1], CPC2, KIN-A, and KIN-B, and truncations thereof (*Figure 3—figure supplement 1A–D*). AF2 confidently predicted parallel coiled coils between KIN-A and KIN-B, with the main region of interaction contained within the central region of KIN-A (residues ~310–550) and the C-terminal region of KIN-B (residues ~320–580). The C-terminal tail (residues ~550–862) of KIN-A is predicted to be intrinsically disordered (pLDDT scores <20, *Figure 3—figure supplement 1A and B*). The first ~55 residues of INCENP[CPC1], predicted to form two α-helices (residues ~7–20

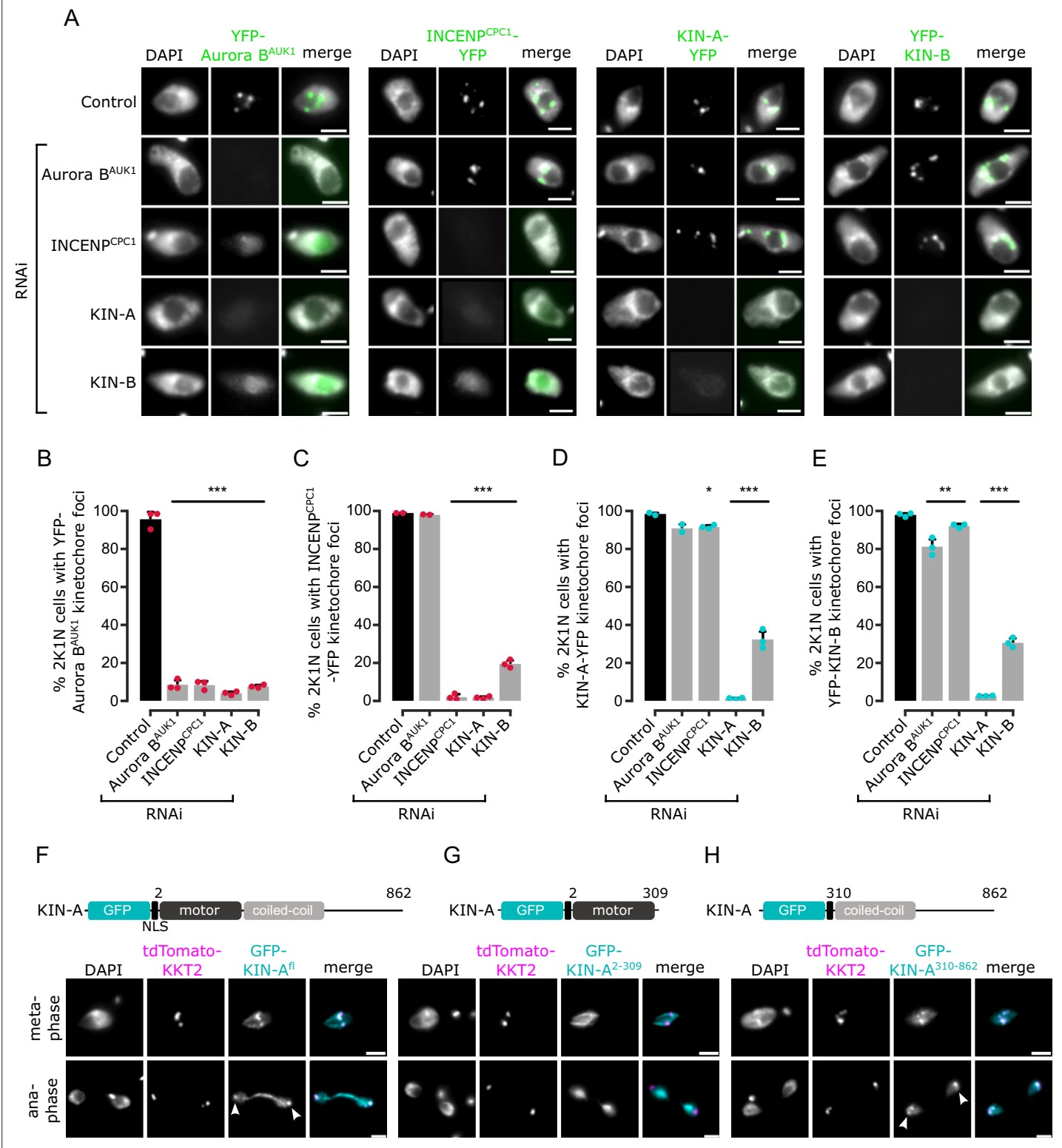

**Figure 2.** Kinetochore localization of the chromosomal passenger complex (CPC) depends on KIN-A and KIN-B. (**A**) Representative fluorescence micrographs showing the localization of YFP-tagged Aurora B[AUK1], INCENP[CPC1], KIN-A, and KIN-B in 2K1N cells upon RNAi-mediated knockdown of indicated CPC subunits. Note that nuclear close-ups are shown here. CPC proteins were not detected in the cytoplasm. RNAi was induced with 1 µg/ml doxycycline for 24 hr (KIN-B RNAi) or 16 hr (all others). Cell lines: BAP3092, BAP2552, BAP2557, BAP3093, BAP2906, BAP2900, BAP2904, BAP3094, BAP2899, BAP2893, BAP2897, BAP3095, BAP3096, BAP2560, BAP2564, BAP3097. Scale bars, 2 µm. (**B–E**) Quantification of 2K1N cells that

*Figure 2 continued on next page*

*Figure 2 continued*

have kinetochore-like dots of YFP-tagged Aurora B[AUK1] (**B**), INCENP[CPC1] (**C**), KIN-A (**D**), and KIN-B (**E**) upon RNAi-mediated depletion of indicated CPC components. All graphs depict the means (bar) ± SD of at least two replicates (shown as dots). A minimum of 100 cells per replicate were quantified. *p<0.05, **p≤0.01, ***p≤0.001 (two-sided, unpaired t-test). (**F–H**) Representative fluorescence micrographs showing the localization of ectopically expressed GFP-KIN-A[fl] (**F**), -KIN-A[2-309] (**G**), and -KIN-A[310-862] (**H**). Expression of GFP fusion proteins was induced with 10 ng/ml doxycycline for 24 hr. Kinetochores are marked with tdTomato-KKT2. Arrowheads indicate KIN-A[fl] and KIN-A[310-862] signals at kinetochores in anaphase. KIN-A[2-309] localizes to the mitotic spindle during (pro)metaphase. Cell lines: BAP2286, BAP2297, BAP2287. Scale bars, 2 μm.

The online version of this article includes the following source data and figure supplement(s) for figure 2:

**Figure supplement 1.** Depletion of chromosomal passenger complex (CPC) proteins causes growth defects and cell cycle arrest.

**Figure supplement 1—source data 1.** Original scans of the western blot analysis (anti-GFP and anti-tubulin) in *Figure 2—figure supplement 1C*.

**Figure supplement 1—source data 2.** PDF containing the uncropped, original scans of the western blot analysis (anti-GFP and anti-tubulin) in *Figure 2—figure supplement 1C* with highlighted bands and sample labels.

and ~36–55), interact with the coiled coils of KIN-A:KIN-B in close proximity to the kinesin motors. A flexible central linker in INCENP[CPC1] bridges the N-terminus of INCENP[CPC1] and the catalytic module of the CPC (Aurora B[AUK1] + INCENP[CPC1] IN-box). Similarly to INCENP[CPC1], an N-terminal α-helical region in CPC2 (residues ~19–75) interacts with the KIN-A:KIN-B coiled coils immediately downstream of the INCENP[CPC1] binding site. Consistent with these predictions, GFP-tagged INCENP[CPC1 2-147] or CPC2[2-120] displayed normal localization dynamics indistinguishable from the corresponding full-length constructs (*Figure 3—figure supplement 1E, F, H, I*). In contrast, deletion of the N-terminal domains of INCENP[CPC1] or CPC2 impaired kinetochore localization (*Figure 3—figure supplement 1G and I*). Together, these data suggest that Aurora B[AUK1] forms a subcomplex with the C-terminus of INCENP[CPC1], and that INCENP[CPC1] and CPC2 interact with the coiled-coil domain of KIN-A:KIN-B via their N-terminal domains.

To validate these findings biochemically, we performed bis(sulfosuccinimidyl) suberate (BS[3]) cross-linking of native CPC complexes isolated by immunoprecipitation of endogenously tagged YFP-Aurora B[AUK1] from cells arrested prior to anaphase, followed by mass spectrometry analysis (IP-CLMS) (*Figure 3A* and *Supplementary file 2*). Indeed, our IP-CLMS data revealed high-score crosslinks between the predicted coiled-coil domains of KIN-A and KIN-B, suggesting that the two kinesins form a parallel heterodimer. As expected, Aurora B[AUK1] formed contacts with the C-terminal IN-box of INCENP[CPC1], consistent with these two proteins constituting the catalytic module of the CPC. The N-terminal region of INCENP[CPC1] interacted mainly with KIN-B and to a lesser degree with KIN-A, with most contacts confined to the N-terminal ends of the predicted coiled-coil domains of the kinesins close to their motor domains. The N-terminus of CPC2, on the other hand, formed crosslinks with the coiled-coil domain of KIN-A. We used PyXlinkViewer (*Schiffrin et al., 2020*) to map our IP-CLMS data onto the assembled AF2 model of the trypanosome CPC (*Figure 3B*). Using a Euclidean distance cut-off of 30 Å, ~85% of crosslinks were compatible with the model, providing confidence in the AF2 predictions. The few crosslinks that violated the distance constraints mainly represent intra-protein contacts between the INCENP[CPC1] N- and C-terminal domains or inter-protein contacts between the INCENP[CPC1] C-terminal domain and the kinesin motor domain of KIN-A. The central domain of INCENP[CPC1] is predicted to be unstructured (pLDDT scores <20, *Figure 3—figure supplement 1C and D*) and may act as a flexible linker, permitting multiple orientations of the catalytic module relative to the KIN-A:KIN-B scaffold. Taken together, these data indicate that KIN-A and KIN-B interact via their coiled-coil domains, which serve as a scaffold for the assembly of the remaining CPC subunits.

## The CPC is recruited to kinetochores through the KKT7-KKT8 complex pathway

Core components of the Haspin-H3T3ph and Bub1-H2AT120ph-Sgo1 pathways that control CPC recruitment to the centromere in other model eukaryotes are not found in kinetoplastids (*Berriman et al., 2005*), and so far, no centromere-specific histone modifications and/or histone variants have been uncovered in *T. brucei*. We reasoned that the centromere receptor(s) of the trypanosome CPC may lie within the repertoire of unconventional kinetochore proteins present in kinetoplastids. Our IP-CLMS approach failed to detect crosslinks between CPC subunits and kinetochore components (*Supplementary file 2*), possibly due to the transient nature of these interactions. Nevertheless,

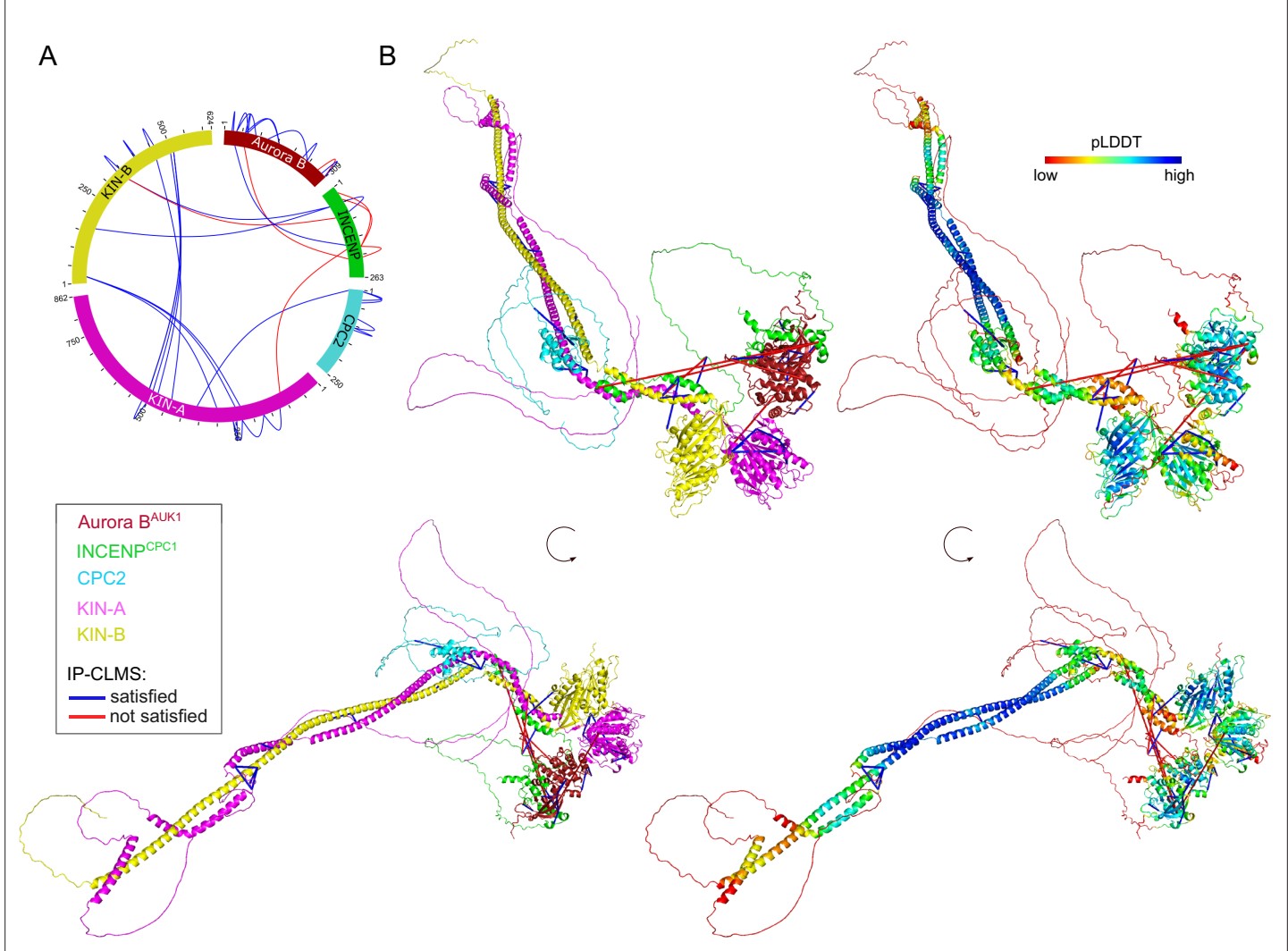

**Figure 3.** Structural model of the trypanosome chromosomal passenger complex (CPC). (**A**) Circular view of the bis(sulfosuccinimidyl) suberate (BS³) crosslinks observed between the subunits of the trypanosome CPC, obtained from native complexes isolated by immunoprecipitation of YFP-Aurora B[AUK1] (cell line: BAP2198). pLink2 (***Chen et al., 2019***) was used to obtain crosslinks from mass spectrometry data. xiView (***Graham et al., 2019***) was used for data visualization. Only crosslinks with a score better than e⁻³ are shown. See ***Supplementary file 2*** for all crosslinks identified by mass spectrometry. (**B**) Cartoon representation showing two orientations of the trypanosome CPC, colored by protein on the left (Aurora B[AUK1]: crimson, INCENP[CPC1]: green, CPC2: cyan, KIN-A: magenta, and KIN-B: yellow) or according to their pLDDT values on the right, assembled from AlphaFold2 predictions shown in ***Figure 3—figure supplement 1***. The pLDDT score is a per-residue estimate of the confidence in the AlphaFold prediction on a scale from 0 to 100. pLDDT >70 (blue, cyan) indicates a reasonable accuracy of the model, while pLDDT <50 (red) indicates a low accuracy and often reflects disordered regions of the protein (***Jumper et al., 2021***). BS³ crosslinks in (**B**) were mapped onto the model using PyXlinkViewer (blue = distance constraints satisfied, red = distance constraints violated, Cα-Cα Euclidean distance threshold = 30 Å) (***Schiffrin et al., 2020***).

The online version of this article includes the following figure supplement(s) for figure 3:

**Figure supplement 1.** AlphaFold2 models of chromosomal passenger complex (CPC) subcomplexes and localization of CPC protein 1 (CPC1) and CPC2 truncations.

several KKT proteins were commonly enriched in the immunoprecipitates of Aurora B[AUK1], INCENP[CPC1], KIN-A, and KIN-B, the most abundant ones being KKT6, KKT7, KKT8, KKT9, KKT10, KKT11, and KKT12 (***Figure 1A***; ***Figure 1—figure supplement 1A***; and ***Supplementary file 2***). KKT7 is detected at kinetochores from S phase until the end of anaphase and recruits the KKT8 complex (comprising KKT8, KKT9, KKT11, and KKT12) (***Akiyoshi and Gull, 2014***; ***Ishii and Akiyoshi, 2020***). The KKT8 complex localizes at kinetochores from S phase and dissociates at the metaphase-anaphase transition.

Using previously validated RNAi constructs (*Akiyoshi and Gull, 2014*; *Llauró et al., 2018*; *Marcianò et al., 2021*), we found that knockdown of KKT7 or KKT9 resulted in dispersal of YFP-Aurora B^AUK1 from kinetochores in ~70% of cells (*Figure 4A–D*). In contrast, depletion of KKT1, KKT2, KKT3, KKT4, KKT6, KKT10/19, KKT14, and KKT16 had little or no effect on YFP-Aurora B^AUK1 localization (*Figure 4—figure supplement 1A and B*). Knockdown of KKT8 complex subunits also impaired kinetochore recruitment of KIN-A-YFP (*Figure 4—figure supplement 1C and D*). We next tested whether KKT7 or the KKT8 complex were able to recruit Aurora B^AUK1 to an ectopic locus using the LacI-LacO system (*Landeira and Navarro, 2007*). For these experiments, we expressed GFP-tagged KKT7^2-261 or KKT8 fused to the Lac repressor (LacI) in trypanosomes containing an ectopic Lac operator (LacO) array stably integrated into rDNA repeats. We previously showed that KKT7 lies upstream of the KKT8 complex (*Ishii and Akiyoshi, 2020*). Indeed, GFP-KKT7^2-261-LacI recruited tdTomato-KKT8, -KKT9, and -KKT12 (*Figure 4—figure supplement 1E*). Expression of GFP-KKT7^2-261-LacI or GFP-KKT8-LacI resulted in robust recruitment of tdTomato-Aurora B^AUK1 to LacO foci in S phase (*Figure 4E and F*). Intriguingly, we also noticed that, unlike endogenous KKT8 (which is not present in anaphase), ectopically expressed GFP-KKT8-LacI remained at kinetochores during anaphase (*Figure 4F*). This resulted in a fraction of tdTomato-Aurora B^AUK1 being trapped at kinetochores during anaphase instead of migrating to the central spindle (*Figure 4F*). We observed a comparable situation upon ectopic expression of GFP-KIN-A, which is retained on anaphase kinetochores together with tdTomato-KKT8 (*Figure 4—figure supplement 1F*). In contrast, Aurora B^AUK1 was not recruited to LacO foci marked by GFP-KKT7^2-261-LacI in anaphase (*Figure 4E*).

## KKT7 recruits the KKT8 complex via the KKT9:KKT11 subcomplex

To gain further insights into the structure and assembly hierarchy within the KKT7-KKT8 complex pathway, we performed CLMS on native complexes isolated by immunoprecipitation of endogenously YFP-tagged kinetochore proteins and mapped the detected crosslinks onto AF2 structure predictions (*Figure 4G and H*; *Supplementary file 2*) (Materials and methods). AF2 confidently predicted coiled coils between KKT8 and KKT12 and between KKT9 and KKT11 (*Figure 4G*; *Figure 4—figure supplement 1G and H*), suggesting that KKT8:KKT12 and KKT9:KKT11 each form distinct subcomplexes. To validate these findings, we co-expressed combinations of 6HIS-KKT8, KKT9, KKT11, and KKT12 in *E. coli* and performed metal affinity chromatography (*Figure 4I*). 6HIS-KKT8 efficiently pulled down KKT9, KKT11, and KKT12, as shown previously (*Ishii and Akiyoshi, 2020*). In the absence of KKT9, 6HIS-KKT8 still pulled down KKT11 and KKT12. Removal of either KKT9 or KKT11 did not impact formation of the KKT8:KKT12 subcomplex. In contrast, 6HIS-KKT8 could not be recovered without KKT12, indicating that KKT12 is required for the formation of the full KKT8 complex. These results support the idea that the KKT8 complex consists of KKT8:KKT12 and KKT9:KKT11 subcomplexes. The two subcomplexes appear to be connected to each other through interactions between the C-terminal region of the KKT8:KKT12 coiled coils and the C-terminus of KKT11. Two alpha helices in KKT7^2-261 (residues ~149–181) are predicted to interact with KKT9:KKT11. Using a distance cut-off of 30 Å, ~70% of crosslinks were compatible with the model (*Figure 4G and H*). Of the crosslinks that failed to meet the distance criteria, ~90% involved unstructured regions within KKT7 or the C-terminal tail of KKT8. Collectively, our results reveal that KKT7 recruits the KKT8 complex through interaction with the KKT9:KKT11 subcomplex.

We next examined the localization dependency of KKT8 complex components in cells. Using RNAi constructs against individual subunits of the KKT8 complex (*Akiyoshi and Gull, 2014*; *Ishii and Akiyoshi, 2020*; *Marcianò et al., 2021*), we assessed localization of endogenously YFP-tagged KKT8, KKT9, and KKT12 (*Figure 4—figure supplement 2A–D*). We found that knockdown of any subunit of the KKT8 complex affected protein levels and kinetochore localization of the other subunits, indicating that presence of all subunits is required to stabilize the full complex. YFP-KKT9 was least affected and was still detectable at kinetochores in ~50% of cells depleted of KKT8, KKT11, or KKT12 (*Figure 4—figure supplement 2D*). Thus, KKT9:KKT11 may lie upstream of KKT8:KKT12. Indeed our IP-CLMS data suggest that the KKT9:KKT11 subcomplex directly interacts with the N-terminus of KKT7 (*Figure 4H* and *Supplementary file 2*). KKT7 also formed robust crosslinks with the KKT10/19 kinases (*Supplementary file 2*), supporting our previous finding that KKT7 and KKT10 form a stable complex (*Ishii and Akiyoshi, 2020*). Together, these data suggest that the KKT7-KKT8 complex pathway serves as the main CPC recruitment arm in trypanosomes.

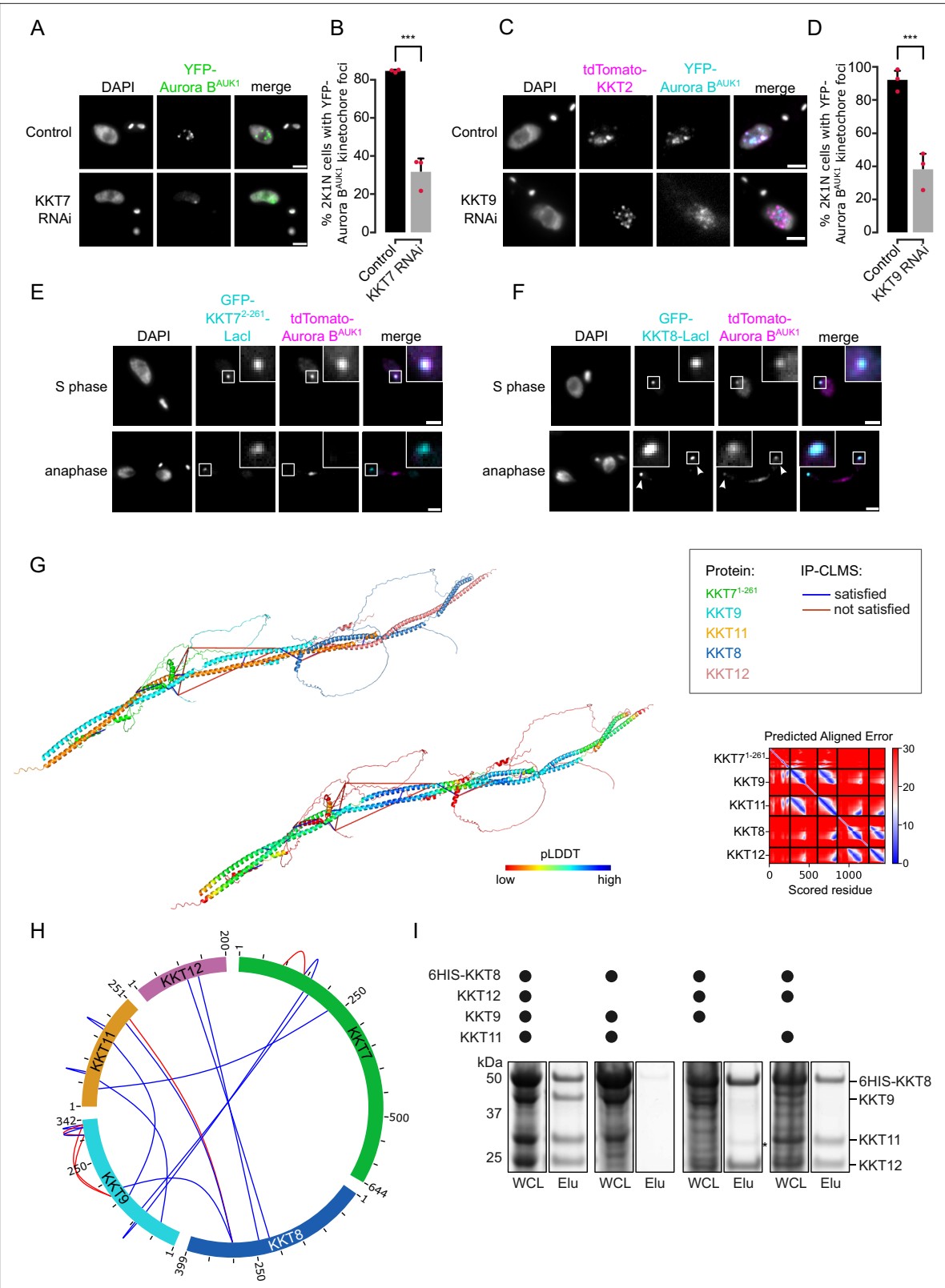

**Figure 4.** The chromosomal passenger complex (CPC) is recruited to kinetochores via the KKT7-KKT8 complex pathway. (**A**) Representative fluorescence micrographs showing the localization of YFP-Aurora B[AUK1] upon RNAi-mediated knockdown of KKT7. RNAi was induced with 1 µg/ml doxycycline for 24 hr. Cell line: BAP577. Scale bars, 2 µm. (**B**) Quantification of 2K1N cells that have kinetochore-like dots of YFP-Aurora B[AUK1] upon knockdown of KKT7. All graphs depict the means (bar) ± SD of three replicates (shown as dots). A minimum of 50 cells per replicate were quantified. ***p≤0.001 (two-

*Figure 4 continued on next page*

*Figure 4 continued*

sided, unpaired t-test). (**C**) Representative fluorescence micrographs showing the localization of YFP-Aurora B[AUK1] upon RNAi-mediated knockdown of KKT9. RNAi was induced with 1 μg/ml doxycycline for 24 hr. Kinetochores are marked with tdTomato-KKT2. Cell line: BAP2276. Scale bars, 2 μm. (**D**) Quantification of 2K1N cells that have kinetochore-like dots of YFP-Aurora B[AUK1] upon knockdown of KKT9. All graphs depict the means (bar) ± SD of three replicates (shown as dots). A minimum of 50 cells per replicate were quantified. ***p≤0.001 (two-sided, unpaired t-test). (**E** and **F**) Representative micrographs of cells in S phase and anaphase showing recruitment of tdTomato-Aurora B[AUK1] to LacO foci marked by ectopically expressed GFP-KKT7[2-261]-LacI (**E**) or -KKT8-LacI (**F**). The insets show the magnification of the boxed region. Expression of LacI fusion proteins was induced with 10 ng/ml doxycycline for 24 hr. Arrowheads in (**F**) indicate anaphase kinetochore localization of GFP-KKT8-LacI and tdTomato-Aurora B[AUK1]. Anaphase kinetochore localization of tdTomato-Aurora B[AUK1] was observed in 75% of anaphase cells expressing GFP-KKT8-LacI (n=28). Cell lines: BAP1395, BAP2640. Scale bars, 2 μm. Of note, LacI fusions with INCENP[CPC1], KIN-A, and KIN-B constructs robustly localized to kinetochores like their endogenous counterparts and failed to form distinct LacI foci and could therefore not be used to assess ectopic recruitment of KKT proteins. (**G**) AlphaFold2 model of the KKT7-KKT8 complex, colored by protein (KKT7[1-261]: green, KKT8: blue, KKT12: pink, KKT9: cyan, and KKT11: orange) (left) and by pLDDT (center). Bis(sulfosuccinimidyl) suberate (BS[3]) crosslinks in (**H**) were mapped onto the model using PyXlinkViewer (*Schiffrin et al., 2020*) (blue = distance constraints satisfied, red = distance constraints violated, Cα-Cα Euclidean distance threshold = 30 Å). Right: Predicted aligned error (PAE) plot of model shown on the left (rank_2). The color indicates AlphaFold's expected position error (blue = low, red = high) at the residue on the x axis if the predicted and true structures were aligned on the residue on the y axis (*Jumper et al., 2021*). (**H**) Circular view of the BS[3] crosslinks observed among KKT7 and KKT8 complex subunits, obtained from native complexes isolated by immunoprecipitation of YFP-tagged KKIP1 (cell line: BAP710). pLink2 (*Chen et al., 2019*) was used to obtain crosslinks from mass spectrometry data and xiView (*Graham et al., 2019*) was used for data visualization. Only crosslinks with a score better than e[−3] are shown. See *Supplementary file 2* for all crosslinks identified by mass spectrometry. (**I**) Indicated combinations of 6HIS-tagged KKT8 (~46 kDa), KKT9 (~39 kDa), KKT11 (~29 kDa), and KKT12 (~23 kDa) were co-expressed in *Escherichia coli*, followed by metal affinity chromatography and SDS-PAGE. The asterisk indicates a common contaminant. Raw, uncropped gels are shown in *Figure 4—source data 1 and 2*.

The online version of this article includes the following source data and figure supplement(s) for figure 4:

**Source data 1.** Original scans of the SimplyBlue-stained SDS-PAGE gel in *Figure 4I*.

**Source data 2.** PDF containing the uncropped, original scans of the SimplyBlue-stained SDS-PAGE gel in *Figure 4I* with highlighted bands and sample labels.

**Figure supplement 1.** Kinetochore localization of Aurora B[AUK1] and KIN-A depends on the KKT7-KKT8 complex pathway.

**Figure supplement 2.** Co-dependencies of KKT8 complex subunits for kinetochore localization.

## The KIN-A C-terminal tail interacts with the KKT8 complex through a conserved domain

By IP-CLMS we failed to detect reliable crosslinks between the CPC and the KKT7-KKT8 complex or other kinetochore proteins (*Supplementary file 2*). This suggests that the IP-CLMS approach, although well suited for characterizing stable protein complexes, may not be sensitive enough to detect transient or lower affinity interactions. To overcome this, we used AF2 to probe for potential interactions between the KKT8 complex and chromosomal passenger subunits or (sub)complexes. AF2 did not predict interactions between the KKT8 complex and INCENP[CPC1], CPC2, or KIN-B (data not shown). Intriguingly, AF2 predicted with high confidence interactions between KKT9:KKT11 and a conserved region (residues ~722–741) within the KIN-A C-terminal tail, which we termed <u>c</u>onserved <u>d</u>omain 1 (CD1) (*Figure 5A and B*; *Figure 5—figure supplements 1 and 2A*). This interaction involves a triple helix composed of KIN-A CD1, KKT9, and KKT11 in a region close to the KKT7-binding site. pLDDT scores improved significantly for KIN-A CD1 in complex with KKT9:KKT11 (>80) compared to KIN-A CD1 alone (~20) (*Figure 3—figure supplement 1A and B*), suggesting that CD1 forms a helical structure upon binding to KKT9:KKT11. Sequence alignment revealed the presence of a second conserved domain (residues ~816–862) within the C-terminal tail of KIN-A, hereafter referred to as CD2 (*Figure 5B*; *Figure 5—figure supplement 1*). To assess the contributions of CD1 and CD2 for kinetochore recruitment of KIN-A in vivo, we ectopically expressed GFP fusions of the central coiled coils and C-terminal tail of KIN-A (residues ~310–862) lacking either CD1, CD2, or both (*Figure 5—figure supplement 2B–E*). GFP-KIN-A[310-862 ΔCD2] showed a moderate reduction in kinetochore localization in metaphase and was completely lost from kinetochores in anaphase (*Figure 5—figure supplement 2C and F*). By contrast, GFP-KIN-A[310-862 ΔCD1] was largely dispersed in metaphase but reappeared at kinetochores in anaphase (*Figure 5—figure supplement 2D and F*). GFP-KIN-A[310-716] lacking both CD1 and CD2 failed to enrich at kinetochores both in metaphase and anaphase (*Figure 5—figure supplement 2E and F*). These data suggest that CD1 and CD2 synergistically promote kinetochore localization of KIN-A, with CD1 interacting with KKT9:KKT11 and CD2 possibly interacting with another receptor at the kinetochore.

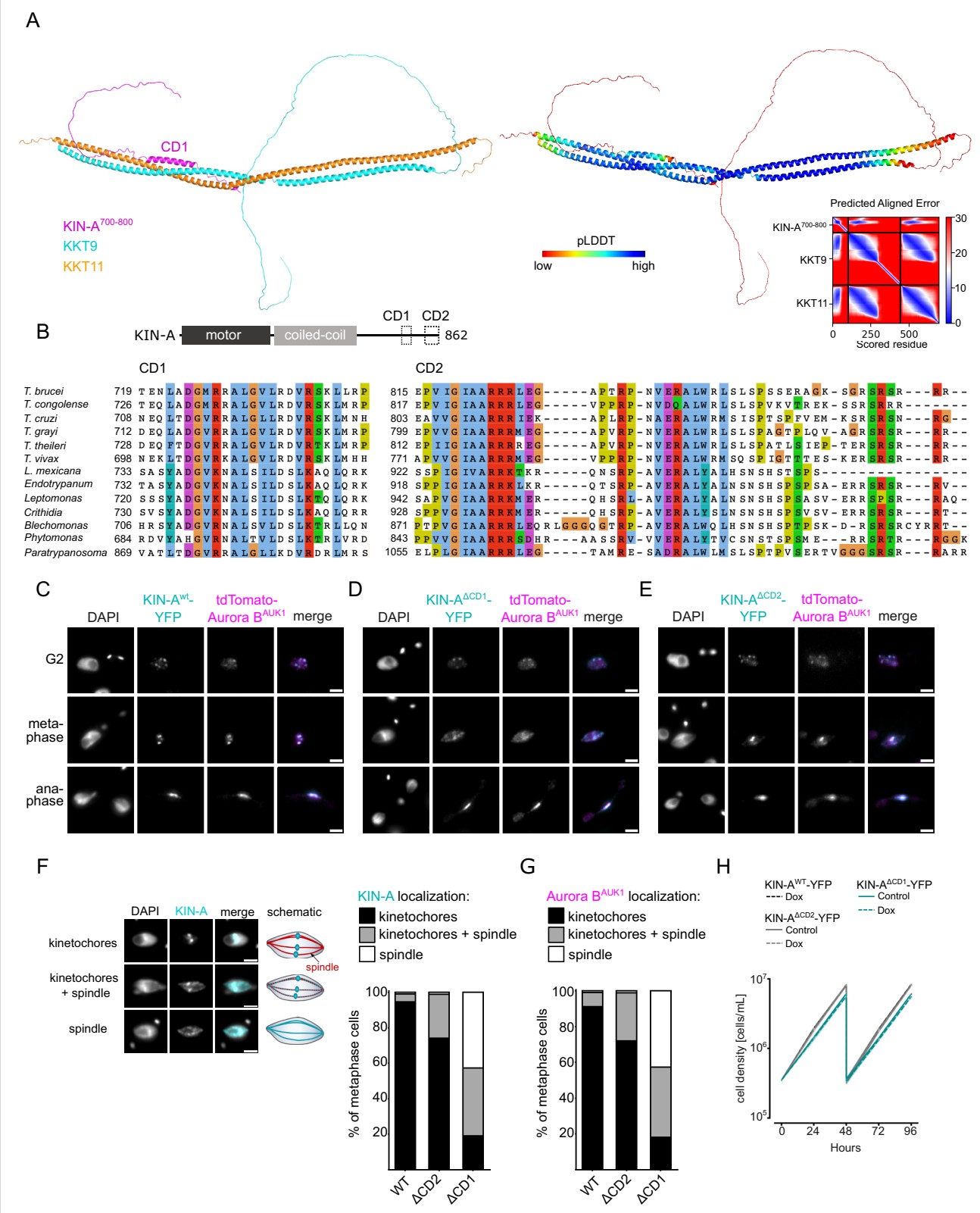

**Figure 5.** Two conserved domains within the C-terminal tail of KIN-A promote kinetochore recruitment of the chromosomal passenger complex (CPC). (**A**) Left: AlphaFold2 model of KKT9:KKT11 in complex with KIN-A$^{700-800}$. Cartoon representations are colored by protein (KKT9: cyan, KKT11: orange, KIN-A: magenta) (left) or according to their pLDDT values (blue = high confidence, red = low confidence) (center). Right: Predicted aligned error (PAE) plot of model (rank_1) predicted by AlphaFold2 (blue = high confidence, red = low confidence in the relative positions of the domains to one another).

*Figure 5 continued on next page*

*Figure 5 continued*

Conserved domain 1 (CD1) of KIN-A was predicted to interact with KKT9:KKT11 in all five AlphaFold2 models (rank_1 to rank_5). (**B**) Multiple sequence alignment of KIN-A CD1 and CD2 showing conservation. (**C–E**) Representative fluorescence micrographs showing the localization of tdTomato-Aurora B$^{AUK1}$ and YFP-tagged KIN-A$^{wt}$ (**C**), KIN-A$^{\Delta CD1 (717–743)}$ (**D**), and KIN-A$^{\Delta CD2 (816–862)}$ (**E**). RNAi was induced with 1 µg/ml doxycycline for 24 hr to deplete the untagged KIN-A allele. Cell lines: BAP3067, BAP3128, BAP3127. Scale bars, 2 µm. (**F**) Stacked bar charts showing the percentage of YFP-tagged KIN-A$^{wt}$, KIN-A$^{\Delta CD1}$, and KIN-A$^{\Delta CD2}$ on kinetochores, kinetochores+spindle, and spindle only in metaphase cells. Examples and schematic illustrations of the three categories used for scoring are presented on the left. A minimum of 50 cells per condition were quantified. (**G**) Stacked bar charts showing the percentage of tdTomato-Aurora B$^{AUK1}$ on kinetochores, kinetochores+spindle, and spindle only in metaphase cells upon rescue with YFP-tagged KIN-A$^{wt}$, KIN-A$^{\Delta CD1}$, or KIN-A$^{\Delta CD2}$. A minimum of 50 cells per condition were quantified. (**H**) Growth curves for indicated cell lines and conditions. RNAi was induced with 1 µg/ml doxycycline for to deplete the untagged KIN-A allele in the knockdown conditions and cultures were diluted at day 2. Cell lines: BAP3067, BAP3128, BAP3127.

The online version of this article includes the following figure supplement(s) for figure 5:

**Figure supplement 1.** Multiple sequence alignment of KIN-A.

**Figure supplement 2.** Conserved domain 1 (CD1) and CD2 contribute synergistically to kinetochore localization of KIN-A.

We next tested the relevance of KIN-A CD1 and CD2 for CPC localization and function by replacing one allele of KIN-A with C-terminally tagged wild-type or mutant constructs lacking either CD1 or CD2 and performed RNAi against the 3'UTR of KIN-A to deplete the untagged allele. We were unable to obtain a rescue cell line lacking both CD1 and CD2 as the double-mutant protein was not properly expressed. Wild-type KIN-A-YFP along with Aurora B$^{AUK1}$ robustly localized to kinetochores from S phase until anaphase onset (*Figure 1C and D*; *Figure 5C*). KIN-A$^{\Delta CD1}$-YFP was detectable at kinetochores in G2 but predominantly localized to the mitotic spindle from (pro)metaphase onward (*Figure 5D and F*), indicating that removal of CD1 severely weakens the affinity of KIN-A for kinetochores and instead shifts the balance toward microtubule binding. Interestingly, expression of KIN-A$^{\Delta CD1}$-YFP similarly affected the localization of Aurora B$^{AUK1}$ (*Figure 5D and G*; *Figure 5—figure supplement 2G*). We also detected partial spindle localization of Aurora B$^{AUK1}$ in a small population (~25%) of metaphase cells expressing KIN-A$^{\Delta CD2}$-YFP (*Figure 5E–G*). Central spindle localization of KIN-A in anaphase was unaffected by deletion of either CD1 or CD2 (*Figure 5D and E*). Remarkably, despite a substantial loss of Aurora B$^{AUK1}$ from kinetochores in metaphase, ΔCD1 cells exhibited normal cell cycle profiles (*Figure 5—figure supplement 2H*) and showed only a modest decrease in proliferation rates (*Figure 5H*). This parallels the situation in budding yeast, in which error-free chromosome segregation can be sustained even when inner centromere localization of Aurora B is largely abolished (*Campbell and Desai, 2013*; *García-Rodríguez et al., 2019*).

## CPC targeting to the central spindle in anaphase depends on KIN-A's ATPase activity

Finally, we asked how translocation of the CPC to the spindle midzone in anaphase is achieved in trypanosomes. In mammalian cells, dephosphorylation of INCENP and the kinesin MKLP2 upon anaphase onset allows formation of a transient CPC-MKLP2 complex (*Gruneberg et al., 2004*; *Hümmer and Mayer, 2009*; *Kitagawa et al., 2014*; *Serena et al., 2020*). MKLP2 activity then drives plus-directed movement of this complex along microtubules of the anaphase spindle. We therefore speculated that anaphase translocation of the kinetoplastid CPC to the central spindle may involve the kinesin motor domain of KIN-A. KIN-B is unlikely to be a functional kinesin based on the absence of several well-conserved residues and motifs within the motor domain, which are fully present in KIN-A (*Li et al., 2008*). These include the P-loop, switch I and switch II motifs, which form the nucleotide-binding cleft, and many conserved residues within the α4-L12 elements, which interact with tubulin (*Figure 6—figure supplement 1*; *Endow et al., 2010*). Consistent with this, the motor domain of KIN-B, contrary to KIN-A, failed to localize to the mitotic spindle when expressed ectopically (*Figure 2—figure supplement 1E*) and did not co-sediment with microtubules in our in vitro assay (*Figure 6—figure supplement 2A*).

Ectopically expressed GFP-KIN-A and -KIN-A$^{2-309}$ partially localized to the mitotic spindle but failed to concentrate at the midzone during anaphase (*Figure 2F and G*), suggesting that N-terminal tagging of the KIN-A motor domain may interfere with its function. To address whether the ATPase activity of KIN-A is required for central spindle localization of the CPC, we replaced one allele of KIN-A with a C-terminally YFP-tagged G210A ATP hydrolysis-defective rigor mutant (*Figure 6A*; *Rice et al., 1999*)

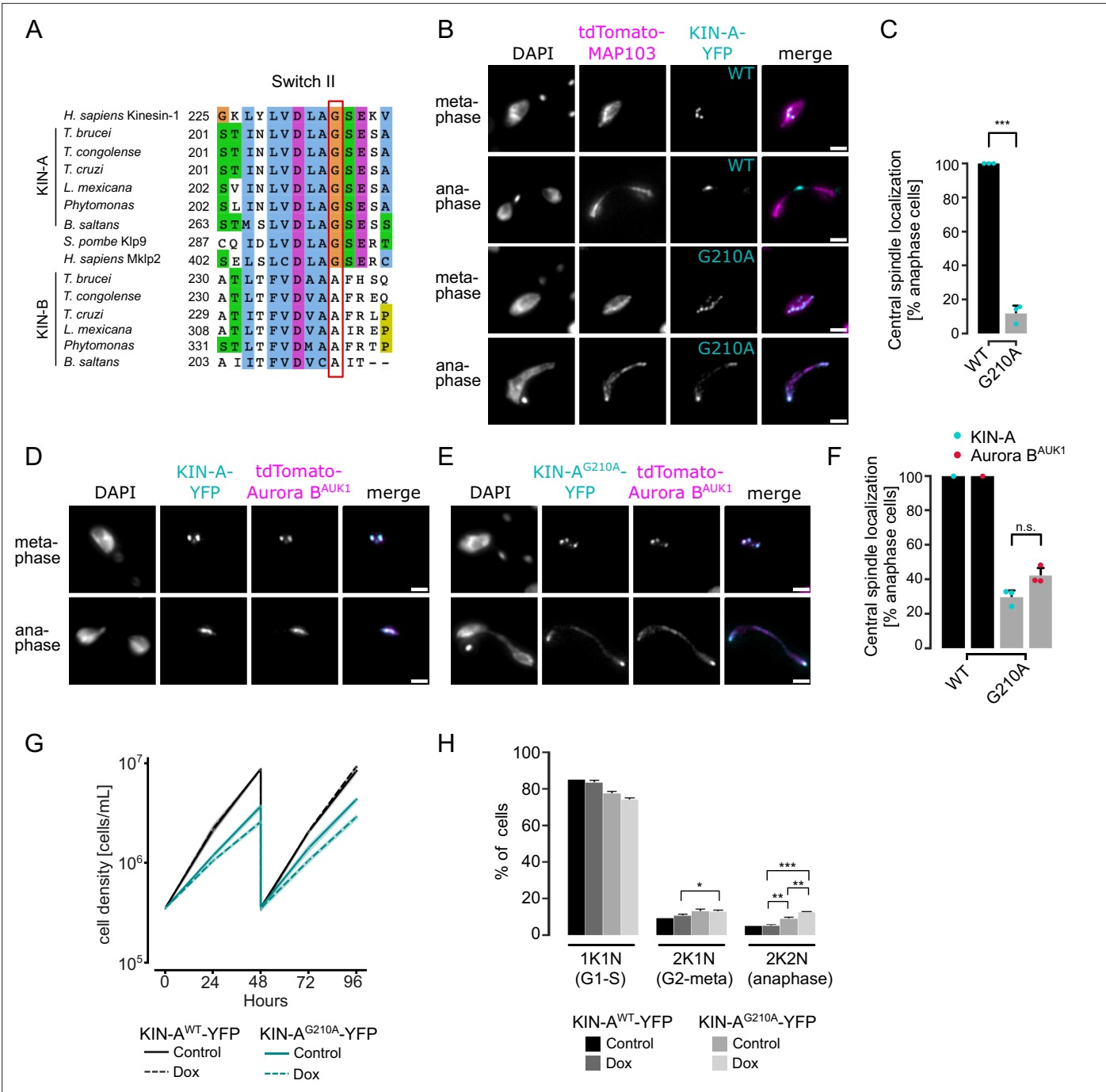

**Figure 6.** KIN-A ATPase activity is required for central spindle localization of the chromosomal passenger complex (CPC) in anaphase. (**A**) Multiple sequence alignment showing conservation of switch II region in KIN-A and KIN-B, with the key glycine residue (G210 in *T. brucei*) targeted for rigor mutation highlighted in red. (**B**) Representative fluorescence micrographs showing the localization of tdTomato-MAP103 (spindle marker) and YFP-tagged KIN-A$^{wt}$ or KIN-A$^{G210A}$ (rigor mutant). RNAi was induced with 1 µg/ml doxycycline for 24 hr to deplete the untagged KIN-A allele. Cell lines: BAP3068, BAP3071. Scale bars, 2 µm. (**C**) Quantification showing the percentage of anaphase cells that have YFP-tagged KIN-A$^{wt}$ or KIN-A$^{G210A}$ localized at the central spindle. All graphs depict the means (bar) ± SD of three replicates (shown as dots). A minimum of 40 cells per replicate were quantified. ***p≤0.001 (two-sided, unpaired t-test). (**D** and **E**) Representative fluorescence micrographs showing the localization of tdTomato-Aurora B$^{AUK1}$ and YFP-tagged KIN-A$^{wt}$ (**D**) or KIN-A$^{G210A}$ (**E**). RNAi was induced with 1 µg/ml doxycycline for 24 hr to deplete the untagged KIN-A allele. Cell lines: BAP3067, BAP3070. Scale bars, 2 µm. (**F**) Quantification showing the percentage of anaphase cells that have tdTomato-Aurora B$^{AUK1}$ localized at the central spindle upon rescue with YFP-tagged KIN-A$^{wt}$ or KIN-A$^{G210A}$. Graphs for the KIN-A$^{G210}$ rescue conditions (gray) depict the means (bar) ± SD of three replicates (shown as dots). A minimum of 30 cells per replicate were quantified. (**G**) Growth curves for indicated cell lines and conditions. RNAi was induced with 1 µg/ml doxycycline for to deplete the untagged KIN-A allele in the knockdown conditions and cultures were diluted at day 2. Cell lines: BAP3064,

*Figure 6 continued on next page*

*Figure 6 continued*

BAP3065. (**H**) Cell cycle profiles for the indicated cell lines and conditions. RNAi was induced with 1 μg/ml doxycycline to deplete the untagged KIN-A allele in the knockdown conditions and cells were fixed after 24 hr. All graphs depict the means (bar) ± SD of at least two replicates. A minimum of 300 cells per replicate were quantified. Cell lines: BAP3064, BAP3065. ***p≤0.001 (two-sided, unpaired t-test).

The online version of this article includes the following source data and figure supplement(s) for figure 6:

**Figure supplement 1.** Multiple sequence alignment of KIN-A and KIN-B from different kinetoplastids with human kinesin-1, human Mklp2, and yeast Klp9.

**Figure supplement 2.** ATPase activity of KIN-A promotes kinetochore alignment at the metaphase plate.

**Figure supplement 2—source data 1.** Original scans of the SimplyBlue-stained SDS-PAGE gel in *Figure 6—figure supplement 2A*.

**Figure supplement 2—source data 2.** PDF containing the uncropped, original scans of the SimplyBlue-stained SDS-PAGE gel in *Figure 6—figure supplement 2A* with highlighted bands and sample labels.

and used an RNAi construct directed against the 3'UTR of KIN-A to deplete the untagged allele. The rigor mutation did not affect recruitment of KIN-A to kinetochores (*Figure 6—figure supplement 2B and C*). However, KIN-A$^{G210A}$-YFP-marked kinetochores were misaligned in ~50% of cells arrested in metaphase, suggesting that ATPase activity of KIN-A promotes chromosome congression to the metaphase plate (*Figure 6—figure supplement 2D–G*). In anaphase, the KIN-A rigor mutant failed to concentrate at the central spindle and instead widely decorated the mitotic spindle, with increased signal observed at spindle poles likely due to poleward flux (*Figure 6B and C*). Importantly, expression of the KIN-A$^{G210A}$ rigor mutant prevented Aurora B$^{AUK1}$ from translocating to the central spindle and caused lagging chromosomes (*Figure 6D–F*). The KIN-A rigor mutation also slowed cell proliferation even in the presence of wild-type protein and caused accumulation of cells in anaphase (*Figure 6G and H*). We conclude that central spindle localization of the CPC depends on KIN-A's ATPase activity and is required for proper chromosome segregation.

## Discussion

Whereas astonishing diversity in kinetochore composition is seen among eukaryotes (*van Hooff et al., 2017*; *Komaki et al., 2022*; *Tromer et al., 2019*), the proteins of the regulatory circuitry underlying chromosome segregation, such as the APC/C, SAC, and CPC, are more widely conserved. Homologs of the CPC proteins Aurora B kinase and its associated partner INCENP have been detected in almost all sequenced eukaryotes, including kinetoplastids. The dynamic localization pattern exhibited by the CPC (e.g. transferring from centromeres to the central spindle upon metaphase-anaphase transition) is likewise highly conserved across eukaryotes but appears to be achieved through a variety of different mechanisms. For instance, while CPC recruitment to the centromeres in higher eukaryotes is governed by two histone phosphorylation marks (Haspin-mediated H3T3ph and Bub1-mediated H2AT120ph), budding yeasts employ a combination of histone modifications (Bub1-mediated H2AT121ph) and kinetochore proteins as CPC receptors. Borealin$^{Bir1}$ not only recognizes Sgo1 but also interacts with the CBF3 complex through Ndc10 (*Cho and Harrison, 2011*; *Yoon and Carbon, 1999*). Furthermore, INCENP$^{Sli15}$/Aurora B$^{Ipl1}$ interact with the Ctf19 subunit of the COMA complex at kinetochores (*Fischböck-Halwachs et al., 2019*). The proteins that form the localization module of the CPC in different species appear to mirror the diversity in centromeric CPC receptors. In fact, many phyla lack Borealin or Survivin homologs (*Komaki et al., 2022*). Komaki et al. recently identified two functionally redundant CPC proteins in *Arabidopsis*, Borealin Related Interactor 1 and 2, which engage in a triple helix bundle with INCENP and Borealin using a conserved helical domain, but employ an FHA domain instead of a BIR domain to read H3T3ph (*Komaki et al., 2022*).

In this study, we have identified KIN-A and KIN-B as components of the CPC in trypanosomes, and delineated a novel pathway for centromeric recruitment of the CPC in this evolutionary divergent group of eukaryotes. In agreement with our work, an early study on the CPC in *T. brucei* found that KIN-A and KIN-B co-purified with Aurora B$^{AUK1}$, INCENP$^{CPC1}$, and CPC2 based on a pull-down of an Aurora B$^{AUK1}$-PTP fusion protein followed by mass spectrometry analysis (*Li et al., 2008*). However, HA-tagged KIN-A and KIN-B were not detected at kinetochores (from late interphase until metaphase) nor at the new FAZ tip (from late anaphase), and hence were not interpreted as CPC proteins. Contrary to this report, our data clearly show that KIN-A and KIN-B are constitutive components of

the CPC in *T. brucei*. First, YFP-tagged KIN-A and KIN-B co-localize with Aurora B[AUK1], INCENP[CPC1], and CPC2 throughout the cell cycle. Second, KIN-A and KIN-B are readily detected within native CPC complexes isolated by immunoprecipitation of CPC subunits from cells arrested prior to anaphase and form robust crosslinks with the other CPC subunits. Finally, YFP-Aurora B[AUK1] and INCENP[CPC1]-YFP are crucially dependent on KIN-A and KIN-B for localizing both to kinetochores from S phase until metaphase and to the central spindle in anaphase. Thus, the KIN-A:KIN-B subcomplex represents the localization module of the trypanosome CPC.

Biochemical, cell biological, and in silico modeling approaches indicate that the kinesins KIN-A and KIN-B form coiled coils between their central and C-terminal domains, respectively, which then serve as a scaffold onto which INCENP[CPC1] and CPC2 assemble via their N-terminal α-helical domains. The catalytic module of the trypanosomes CPC, consisting of Aurora B[AUK1] bound to the C-terminal IN-box of INCENP[CPC1], is connected to the KIN-A:KIN-B scaffold through a flexible linker in INCENP[CPC1]. While our on-beads crosslinking of native CPCs suggests that the catalytic module is positioned in close proximity to the kinesin heads, this may not necessarily be true in vivo. For example, the catalytic module may exist in both 'locked' and 'open' conformations with regard to its association with the kinesin motor domains. We speculate that the interaction of the KIN-A motor domain with microtubules from prometaphase onward may cause the catalytic module to disengage from its kinesin head-associated state (*Figure 7*). In analogy to the SAH domain of INCENP in other model eukaryotes which has been proposed to function as a dog leash (*Samejima et al., 2015*; *Santaguida and Musacchio, 2009*), the INCENP[CPC1] flexible linker in trypanosomes (~100 amino acids long) may then permit Aurora B[AUK1] to roam across a larger but nevertheless spatially constrained target area to phosphorylate its substrates while still being anchored to the kinetochore via KIN-A:KIN-B, which can interact both with kinetochore components and spindle microtubules. Importantly, this mechanism would allow the CPC to act as an intrinsic 'sensor' of KT-MT attachments. Such models dealing with alternative conformations of the CPC in various cellular context will require further testing in the future.

We propose that multiple weak interactions of the KIN-A C-terminal unstructured tail with kinetochore components act in synergy to stabilize the CPC at kinetochores. Critically, these interactions need to be of transient and reversible nature to permit the dynamic release of the CPC upon anaphase onset. Three mechanisms are likely to play a role in this context (*Figure 7*). First, removal of the KKT8 complex (the 'CD1 receptor') at the metaphase-anaphase transition effectively eliminates one of the key CPC-kinetochore interfaces. Second, the affinity of the KIN-A C-terminal tail for its binding partners at the kinetochore may be further fine-tuned through reversible post-translational modifications. In support of this, the KIN-A C-terminal tail harbors many putative CRK3 sites (10 sites matching the minimal S/T-P consensus motif for CDKs) and is also heavily phosphorylated by Aurora B[AUK1] in vitro (*Ballmer et al., 2024*). Finally, we speculate that the interaction of the KIN-A motor domain with microtubules, coupled to the force generating ATP hydrolysis and possibly plus-end-directed motion, eventually outcompetes the weakened interactions of the CPC with the kinetochore and facilitates the extraction of the CPC from chromosomes onto spindle microtubules during anaphase. Indeed, deletion of the KIN-A motor domain or impairment of its motor function through N-terminal GFP tagging causes the CPC to be trapped at kinetochores in anaphase. Central spindle localization is additionally dependent on the ATPase activity of the KIN-A motor domain as illustrated by the KIN-A rigor mutant.

It remains to be investigated whether KIN-A functions as a plus-end-directed motor. The role of the KIN-B in this context is equally unclear. Because KIN-B does not possess a functional kinesin motor domain, we deem it unlikely that the KIN-A:KIN-B heterodimer moves hand-over-hand along microtubules as do conventional (kinesin-1 family) kinesins. Rather, the KIN-A motor domain may function as a single-headed unit and drive processive plus-end-directed motion using a mechanism similar to the kinesin-3 family kinesin KIF1A (*Okada and Hirokawa, 1999*). Formation of transient complexes between the CPC and kinesins or other microtubule plus-end tracking proteins upon anaphase onset appears to be a common theme underlying the central spindle translocation of the CPC. For instance, the CPC interacts with MKLP2 or Bim1[EB1] in human and yeast cells, respectively (*Gruneberg et al., 2004*; *Zimniak et al., 2012*). Thus, a deeper understanding of CPC regulation in trypanosomes is bound to provide evolutionary insights into fundamental principles of chromosome segregation in eukaryotes and can lead to the discovery of druggable targets to combat kinetoplastid diseases (*Saldivia et al., 2020*).

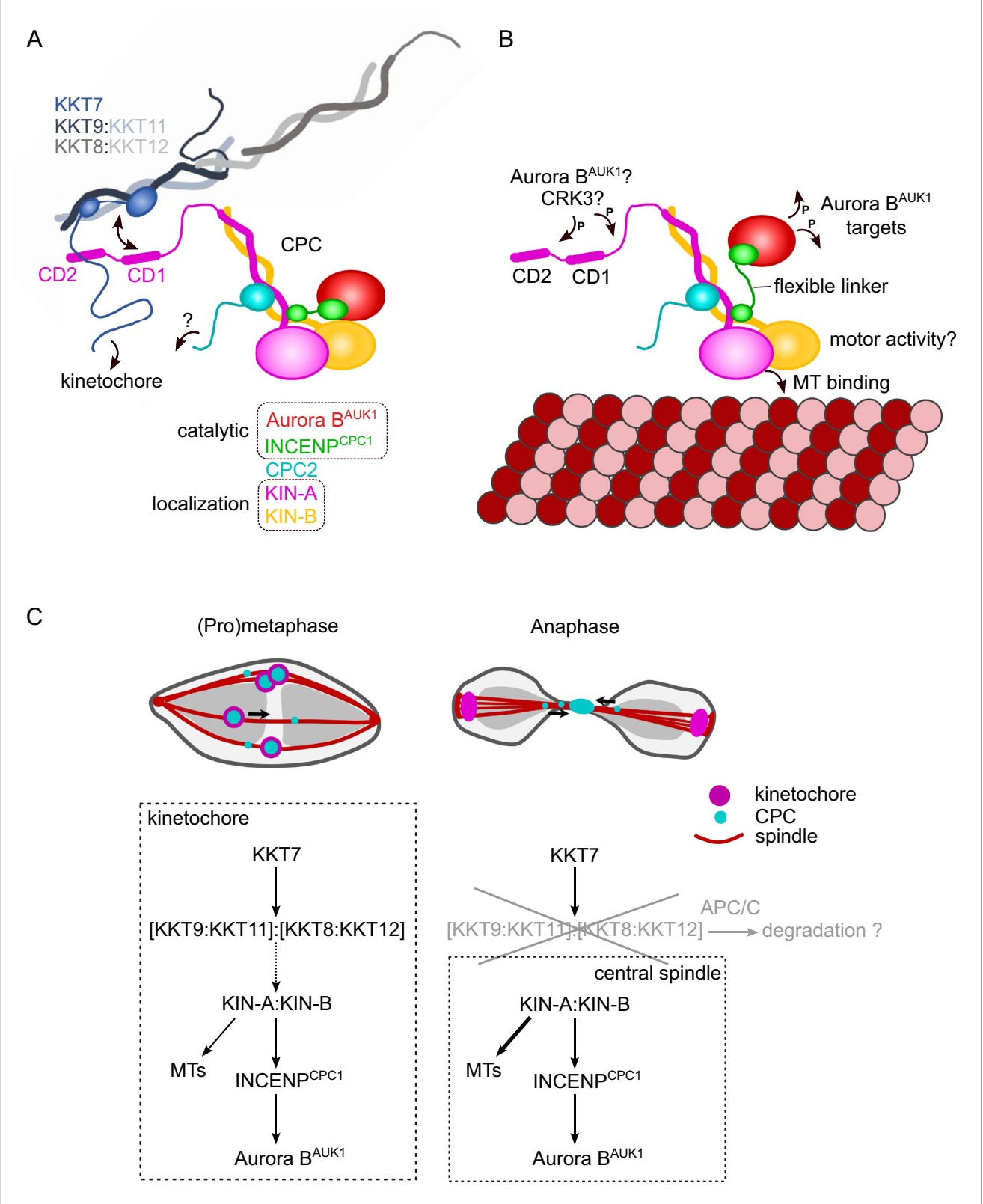

**Figure 7.** Model for chromosomal passenger complex (CPC) localization and function in trypanosomes. (**A**) KIN-A (magenta) and KIN-B (yellow) interact via their coiled-coil domains and form a scaffold for the assembly of CPC2 (cyan) and the catalytic module of the CPC, composed of Aurora B[AUK1] (red) and INCENP[CPC1] (green). During interphase, the catalytic module is positioned close to the kinesin head domains of KIN-A and KIN-B. CPC recruitment to the inner kinetochore is mediated through multiple weak interactions between the C-terminal unstructured tail of KIN-A, containing CD1

*Figure 7 continued on next page*

*Figure 7 continued*

and CD2, with the coiled-coil domain of KKT9:KKT11 (dark blue:light blue) and possibly the N-terminus of KKT7 (blue). The KKT8 complex, comprising KKT9:KKT11 and KKT8:KKT12 (dark gray:light gray) subcomplexes, is connected to other kinetochore proteins through KKT7. Additional kinetochore targeting domains of the CPC may reside within the C-terminus of KIN-B and/or CPC2. We propose that the KIN-A:KIN-B subcomplex represents the main localization module of the trypanosome CPC. As illustrated in (**B**), the affinity of the KIN-A C-terminal tail for its receptor(s) at the kinetochore may be further modulated through phosphorylation by the CDK1 homolog CRK3 and the Aurora B$^{AUK1}$ kinase itself (*Ballmer et al., 2024*). Interaction of the N-terminal motor domain of KIN-A with spindle microtubules (MTs) from prometaphase onward causes the catalytic module to disengage from its kinesin head-associated state. The ~100 amino acid long flexible linker within INCENP$^{CPC1}$ would then permit Aurora B$^{AUK1}$ to phosphorylate its substrates within a larger but nevertheless spatially constrained target area while still being anchored to the kinetochore via KIN-A:KIN-B. The motor domain of KIN-A could thus act as built-in sensor for KT-MT attachments. (**C**) We propose that the trypanosome CPC is recruited to kinetochores via the KKT7-KKT8 complex pathway (dashed arrow) and that motor activity of KIN-A promotes congression of kinetochores to the metaphase plate during early mitosis. The KKT8 complex dissociates from kinetochores at the metaphase-to-anaphase transition and is possibly degraded in an APC/C-dependent manner. Elimination of the KKT8 complex, the primary kinetochore receptor of the CPC, coupled to microtubule (MT) binding and motor activity of the KIN-A motor domain strips the CPC off kinetochores and facilitates its translocation to the central spindle in anaphase.

## Materials and methods
### Cloning
All primers, plasmids, bacmids, and synthetic DNA used in this study as well as their source or construction details are described in *Supplementary file 1*. All constructs were sequence verified.

### Trypanosome culture
All trypanosome cell lines used in this study were derived from *T. brucei* SmOxP927 procyclic form cells (TREU 927/4 expressing T7 RNA polymerase and the tetracycline repressor to allow inducible expression; *Poon et al., 2012*) or from PCF1339 procyclic form cells (TREU 927/4 expressing T7 RNA polymerase, tetracycline repressor, and the Cas9 nuclease; *Beneke et al., 2017*) and are described in *Supplementary file 1*. Cell lines were confirmed by mass spectrometry analysis to be *T. brucei* TREU 927/4. Cell lines with tagged proteins were confirmed by IP-MS and/or microscopy. RNAi-mediated depletion of target proteins was assessed by microscopy. Point mutants were confirmed by sequencing the targeted gene. The absence of *Mycoplasma* contamination was confirmed by DAPI staining. Cells were grown at 28°C in SDM-79 medium supplemented with 10% (vol/vol) heat-inactivated fetal calf serum, 7.5 μg/ml hemin (*Brun and Schönenberger, 1979*), and appropriate selection drugs. Cell growth was monitored using a CASY cell counter (Roche). PCR products or plasmids linearized by NotI were transfected into cells by electroporation (Bio-Rad). Transfected cells were selected by the addition of 30 μg/ml G418 (Sigma), 25 μg/ml hygromycin (Sigma), 5 μg/ml phleomycin (Sigma), or 10 μg/ml blasticidin S (Insight Biotechnology). To obtain endogenously tagged clonal strains, transfected cells were selected by the addition of appropriate drugs and cloned by dispensing dilutions into 96-well plates. Endogenous YFP tagging was performed using the pEnT5-Y vector (*Kelly et al., 2007*) or a PCR-based method (*Dean et al., 2015*). Endogenous tdTomato tagging was performed using pBA148 (*Akiyoshi and Gull, 2014*) and its derivatives. All constructs for ectopic expression of GFP fusion proteins include a short nuclear localization signal (NLS) (*Marchetti et al., 2000*). For doxycycline inducible expression of head-to-head (pBA3-based) and hairpin (pBA310-based) RNAi constructs, GFP-NLS (pBA310-based) and GFP-NLS-LacI fusion proteins (pBA795-based), the linearized plasmids were integrated into 177 bp repeats on minichromosomes. Expression of GFP-NLS or GFP-NLS-LacI fusions was induced by the addition of 10 ng/ml doxycycline for 24 hr. RNAi was induced by the addition of 1 μg/ml doxycycline. LacO-LacI tethering experiments were carried out as described previously using the LacO array inserted at the rDNA locus (*Ishii and Akiyoshi, 2020*; *Landeira and Navarro, 2007*).

### Immunoprecipitation followed by mass spectrometry
For standard immunoprecipitations, 400 ml cultures of asynchronously growing cells were grown to ~5–10 million cells/ml. Cells were pelleted by centrifugation (800×*g*, 10 min), washed once with PBS, and extracted in PEME (100 mM PIPES-NaOH, pH 6.9, 2 mM EGTA, 1 mM MgSO$_4$, and 0.1 mM EDTA) with 1% NP-40, protease inhibitors (10 μg/ml leupeptin, 10 μg/ml pepstatin, 10 μg/ml E-64, and 0.2 mM PMSF) and phosphatase inhibitors (1 mM sodium pyrophosphate, 2 mM Na-β-glycerophosphate, 0.1 mM Na$_3$VO$_4$, 5 mM NaF, and 100 nM microcystin-LR) for 5 min at room temperature

(RT), followed by centrifugation at 1800×*g* for 15 min. Samples were kept on ice from this point on. The pelleted fraction containing kinetochore proteins was resuspended in modified buffer H (BH0.15: 25 mM HEPES, pH 8.0, 2 mM MgCl$_2$, 0.1 mM EDTA, pH 8.0, 0.5 mM EGTA, pH 8.0, 1% NP-40, 150 mM KCl, and 15% glycerol) with protease and phosphatase inhibitors. Samples were sonicated to solubilize kinetochore proteins (12 s, three times with 1 min intervals on ice). 12 µg of mouse monoclonal anti-GFP antibodies (11814460001; Roche) preconjugated with 60 µl slurry of Protein-G magnetic beads (10004D; Thermo Fisher Scientific) with dimethyl pimelimidate (*Unnikrishnan et al., 2012*) were incubated with the extracts for 2.5 hr with constant rotation, followed by four washes with modified BH0.15 containing protease inhibitors, phosphatase inhibitors, and 2 mM DTT. Beads were further washed three times with pre-elution buffer (50 mM Tris-HCl, pH 8.3, 75 mM KCl, and 1 mM EGTA). Bound proteins were eluted from the beads by agitation in 60 µl of elution buffer (0.1% RapiGest [186001860; Waters] and 50 mM Tris-HCl, pH 8.3) for 25 min at RT. Eluates were then incubated at 100°C for 5 min. Proteins were reduced with 5 mM DTT at 37°C for 30 min and alkylated with 10 mM iodoacetamide at 37°C for 30 min. The reaction was quenched by adding 10 mM DTT at 37°C for 30 min, and 100 µl of 20 mM Tris-HCl (pH 8.3) was added. Proteins were digested overnight at 37°C with 0.2 µg trypsin (Promega). Formic acid was then added to 2% and the samples were incubated at 37°C for 30 min to cleave RapiGest, followed by centrifugation for 10 min. The supernatant was desalted over a C18 column and analyzed by electrospray tandem mass spectrometry (MS/MS) over a 60 min gradient using Q-Exactive (Thermo Fisher Scientific) at the Advanced Proteomics Facility (University of Oxford). Peptides were identified by searching MS/MS spectra against the *T. brucei* protein database with MaxQuant (version 2.0.1) with carbamidomethyl cysteine set as a fixed modification and oxidization (Met), phosphorylation (Ser, Thr, and Tyr), and acetylation (Lys) set as variable modifications. Up to two missed cleavages were allowed. The first peptide tolerance was set to 10 ppm. Results were filtered to remove contaminants and reverse hits. Proteins identified with at least two peptides were considered as significant and shown in *Supplementary file 2* (protein FDR 1%).

## Ex vivo crosslinking of the native CPC and kinetochore complexes (IP-CLMS)

For crosslinking IP-MS experiments, cell cultures were scaled up to 1600 ml. Cell cultures were treated with 10 µM MG132 for 4 hr to enrich for cells in metaphase. Cell lysis and immunoprecipitation steps were carried out as described above. After four washes with modified BH0.15 containing protease inhibitors, phosphatase inhibitors and 2 mM DTT, beads were washed three times with 25 mM HEPES pH 7.5, 150 mM NaCl. Proteins were then crosslinked on beads with 0.4 mM BS$^3$ (bis(sulfosuccinimidyl)suberate) (Thermo Fisher Scientific) for 30 min at RT with agitation, followed by three washes in 25 mM HEPES pH 7.5, 150 mM NaCl and a further three washes in 0.1 M ammonium bicarbonate. Samples were then incubated in 8 M urea dissolved in 0.1 M ammonium bicarbonate for 10 min at RT with agitation. Proteins were reduced with 10 mM TCEP for 20 min and alkylated with 10 mM iodoacetamide for 40 min at RT. Proteins were then pre-digested with 0.4 µg LysC for 2 hr at 37°C. The urea concentration in the sample was brought down to <1 M by the addition of 0.1 M ammonium bicarbonate before adding CaCl$_2$ (to 2 mM) and 0.7 µg of trypsin for overnight digestion at 37°C. Formic acid was then added to 2% and the samples were frozen. The crosslinked samples were further processed and analyzed at the proteomics core facility at EMBL Heidelberg. Digested peptides were concentrated and desalted using an OASIS HLB µElution Plate (Waters) according to the manufacturer's instructions. Crosslinked peptides were enriched using size exclusion chromatography (*Leitner et al., 2012*). In brief, desalted peptides were reconstituted with SEC buffer (30% [vol/vol] ACN in 0.1% [vol/vol] TFA) and fractionated using a Superdex Peptide PC 3.2/30 column (GE) on a 1200 Infinity HPLC system (Agilent) at a flow rate of 0.05 ml/min. Fractions eluting between 50 and 70 µl were evaporated to dryness and reconstituted in 30 µl 4% (vol/vol) ACN in 1% (vol/vol) FA. Collected fractions were analyzed by liquid chromatography (LC)-coupled MS/MS using an UltiMate 3000 RSLC nano LC system (Dionex) fitted with a trapping cartridge (µ-Precolumn C18 PepMap 100, 5 µm, 300 µm ID × 5 mm, 100 Å) and an analytical column (nanoEase M/Z HSS T3 column 75 µm × 250 mm C18, 1.8 µm, 100 Å, Waters). Trapping was carried out with a constant flow of trapping solvent (0.05% trifluoroacetic acid in water) at 30 µl/min onto the trapping column for 6 min. Subsequently, peptides were eluted and separated on the analytical column using a gradient composed of solvent A (3% DMSO, 0.1% formic acid in water) and solvent B (3% DMSO, 0.1% formic acid in

acetonitrile) with a constant flow of 0.3 µl/min. The outlet of the analytical column was coupled directly to an Orbitrap Fusion Lumos (Thermo Scientific, San Jose) mass spectrometer using the nanoFlex source. The peptides were introduced into the Orbitrap Fusion Lumos via a Pico-Tip Emitter 360 µm OD × 20 µm ID; 10 µm tip (CoAnn Technologies) and an applied spray voltage of 2.1 kV, instrument was operated in positive mode. The capillary temperature was set at 275°C. Only charge states of 4–8 were included. The dynamic exclusion was set to 30 s and the intensity threshold was $5e^4$. Full mass scans were acquired for a mass range 350–1700 m/z in profile mode in the Orbitrap with resolution of 120,000. The AGC target was set to Standard and the injection time mode was set to Auto. The instrument was operated in data-dependent acquisition mode with a cycle time of 3 s between master scans and MS/MS scans were acquired in the Orbitrap with a resolution of 30,000, with a fill time of up to 100 ms and a limitation of $2e^5$ ions (AGC target). A normalized collision energy of 32 was applied. MS2 data was acquired in profile mode. RAW MS files were searched by the pLink2 software (*Chen et al., 2019*), with carbamidomethyl cysteine set as a fixed and oxidization (Met) set as variable modifications. Up to two missed cleavages were allowed. Precursor tolerance was set to 10 ppm. All the identified crosslinks are shown in *Supplementary file 2* (FDR 5%). Crosslinks were plotted using xiView (*Graham et al., 2019*). All raw mass spectrometry files and custom database files used in this study have been deposited with the ProteomeXchange Consortium via the PRIDE partner repository (*Deutsch et al., 2020*; *Perez-Riverol et al., 2019*) with the dataset identifier PXD045987.

## Fluorescence microscopy

Cells were washed once with PBS, settled onto glass slides, and fixed with 4% paraformaldehyde in PBS for 5 min as described previously (*Nerusheva and Akiyoshi, 2016*). Cells were then permeabilized with 0.1% NP-40 in PBS for 5 min and embedded in mounting media (1% wt/vol 1,4-diazabicyclo[2.2.2] octane, 90% glycerol, 50 mM sodium phosphate, pH 8.0) containing 100 ng/ml DAPI. Images were captured on a Zeiss Axioimager.Z2 microscope (Zeiss) installed with ZEN using a Hamamatsu ORCA-Flash4.0 camera with 63× objective lenses (1.40 NA). Typically, ~20 optical slices spaced 0.2 or 0.24 µm apart were collected. Images were analyzed in ImageJ/Fiji (*Schneider et al., 2012*). Kinetochore localization of endogenously tagged kinetochore proteins or ectopically expressed constructs were examined manually by quantifying the number of cells that clearly had detectable kinetochore-like dots at indicated cell cycle stages. Shown images are central slices.

## In silico structure and interaction predictions

Structures and interactions were predicted with AlphaFold2-Multimer-v2 (*Evans et al., 2022*; *Jumper et al., 2021*) through ColabFold using MMseqs2 (UniRef+Environmental) and default settings (*Mirdita et al., 2022*). Structural predictions of KIN-A/KIN-B, KIN-A$^{310-862}$/KIN-B$^{317-624}$, CPC1/CPC2/ KIN-A$^{300-599}$/KIN-B 317–624, and KIN-A$^{700-800}$/KKT9/KKT11 were performed using ColabFold version 1.3.0 (AlphaFold-Multimer version 2), while those of AUK1/CPC1/CPC2/KIN-A$^{1-599}$/KIN-B, KKT7$^{1-261}$/ KKT9/KKT11/KKT8/KKT12, KKT9/KKT11/KKT8/KKT12, and KKT7$^{1-261}$/KKT9/KKT11 were performed using ColabFold version 1.5.3 (AlphaFold-Multimer version 2.3.1) using default settings, accessed via https://colab.research.google.com/github/sokrypton/ColabFold/blob/v1.3.0/AlphaFold2.ipynb and https://colab.research.google.com/github/sokrypton/ColabFold/blob/v1.5.3/AlphaFold2.ipynb. All structure figures were made using PyMOL version 2.5.2 (Schrödinger, LLC). The following command was used to map pLDDT score onto the AF2 predicted structure models: spectrum b, rainbow_rev, maximum = 100, minimum = 50.

## Protein purification from *E. coli*

Recombinant 6HIS-KKT8, KKT9, KKT11, KKT12 (pBA457) and derivatives were expressed in Rosetta 2(DE3)pLys *E. coli* cells (Novagen). 6HIS-KIN-A$^{2-309}$ (pBA2519) and 6HIS-KIN-B$^{2-316}$ (pBA2513) were expressed in BL21(DE3) cells. Proteins were purified and eluted from TALON beads as previously described (*Llauró et al., 2018*). Briefly, cells were grown in 2xTY media at 37°C to an OD$_{600}$ of ~0.8, at which point protein expression was induced by 0.1 mM IPTG, and then incubated overnight at 20°C. Cells were pelleted at 3400×*g* at 4°C and pellets were resuspended in P500 buffer (50 mM sodium phosphate, pH 7.5, 500 mM NaCl, 5 mM imidazole, and 10% glycerol) supplemented with protease inhibitors (20 µg/ml leupeptin, 20 µg/ml pepstatin, 20 µg/ml E-64, 0.4 mM PMSF) and 1 mM TCEP, and were sonicated on ice. Lysates were treated with benzonase nuclease (500 U/1 l culture)

and spun at 48,000×*g* at 4°C for 30 min. Supernatant was incubated with TALON beads (Takara Clontech) for 1 hr at 4°C, rotating. The beads were washed three times with lysis buffer and proteins were then eluted with elution buffer (P500 buffer containing 250 mM imidazole with 1 mM TCEP). For microtubule co-sedimentation assays, 6HIS-KIN-A$^{2-309}$ and 6HIS-KIN-B$^{2-316}$ were buffer exchanged into BRB80 (80 mM PIPES-KOH, pH 6.9, 1 mM EGTA, and 1 mM MgCl$_2$) with 100 mM KCl using Zeba columns (Thermo Fisher) and flash-frozen in liquid nitrogen for –80°C storage. Polyacrylamide gels were stained with SimplyBlue SafeStain (Invitrogen).

## Microtubule co-sedimentation assay

Microtubule co-sedimentation assays were performed as described previously (*Ludzia et al., 2021*). Briefly, taxol-stabilized microtubules were prepared by mixing 2.5 ml of 100 µM porcine tubulin (Cyto-skeleton) resuspended in BRB80 with 1 mM GTP (Cytoskeleton), 1.25 µl BRB80, 0.5 µl of 40 mM MgCl$_2$, 0.5 µl of 10 mM GTP, and 0.25 µl DMSO, and incubated for 20 min at 37°C. 120 ml of pre-warmed BRB80 containing 12.5 µM Taxol (paclitaxel; Sigma) was added to the sample to bring the microtubule concentration to ~2 µM. 20 µl of 6HIS-KIN-A$^{2-309}$ or 6HIS-KIN-B$^{2-316}$ (at final concentration of 4 µM) in BRB80 with 100 mM KCl were mixed with 20 µl of microtubules (final, 1 µM) and incubated for 45 min at RT. As a control, we incubated 6HIS-KIN-A$^{2-309}$ or 6HIS-KIN-B$^{2-316}$ with BRB80 (with 12.5 µM Taxol). The samples were spun at 20,000×*g* at RT for 10 min, and the supernatant was collected. To the tubes containing pelleted fractions, we added 40 µl of chilled BRB80 with 5 mM CaCl$_2$ and incubated on ice for 5 min to depolymerize microtubules. Following the incubation, samples were boiled for 5 min before SDS-PAGE. Gels were stained with SimplyBlue Safe Stain (Invitrogen). Co-sedimentation assays were performed at least twice with similar results.

## Immunoblotting

Cells were harvested by centrifugation (800×*g*, 5 min) and washed with 1 ml PBS. The pellet was resuspended in 1× LDS sample buffer (Thermo Fisher) with 0.1 M DTT. Denaturation of proteins was performed for 5 min at 95°C. SDS-PAGE and immunoblots were performed by standard methods using the following antibodies: rabbit polyclonal anti-GFP (TP401, 1:5000) and mouse monoclonal TAT1 (anti-trypanosomal-alpha-tubulin, 1:5000, a kind gift from Keith Gull) (*Woods et al., 1989*). Secondary antibodies used were: IRDye 680RD goat anti-mouse (LI-COR, 926-68070) and IRDye 800CW goat anti-rabbit (LI-COR, 926-32211). Bands were visualized on an ODYSSEY Fc Imaging System (LI-COR).

## Multiple sequence alignment

Protein sequences and accession numbers for Aurora B$^{AUK1}$, INCENP$^{CPC1}$, CPC2, KIN-A, and KIN-B used in this study were retrieved from the TriTryp database (*Aslett et al., 2010*), UniProt (*Bateman, 2019*), or a published study (*Butenko et al., 2020*). Searches for homologous proteins were done using BLAST in the TriTryp database (*Aslett et al., 2010*) or using hmmsearch using manually prepared hmm profiles (HMMER, version 3.0; *Eddy, 1998*). The top hit in each organism was considered as a true ortholog only if the reciprocal BLAST search returned the query protein as a top hit in *T. brucei*. Multiple sequence alignment was performed with MAFFT (L-INS-i method, version 7) (*Katoh et al., 2019*) and visualized with the Clustalx coloring scheme in Jalview (version 2.10) (*Waterhouse et al., 2009*).

# Acknowledgements

We thank Miguel Navarro (Instituto de Parasitologıá y Biomedicina López-Neyra, Consejo Superior de Investigaciones Cientificas, Spain) for providing pMig75 and pMig96 plasmids and Midori Ishii Kanazawa and William Carter for helping trypanosome strain construction. We thank the Micron Advanced Bioimaging Unit and the Advanced Proteomics Facility at the University of Oxford, and the Proteomics Core Facility at the EMBL in Heidelberg, especially Mandy Rettel and Jennifer Schwarz, for their support. We also thank Patryk Ludzia and Midori Ishii Kanazawa for comments on our manu-script. D Ballmer was supported by the Berrow Foundation. B Akiyoshi was supported by a Wellcome Trust Senior Research Fellowship (210622/Z/18/Z), a Wellcome Discovery Award (227243/Z/23/Z), and a Centre Core Grant to the Wellcome Trust Centre for Cell Biology (203149).

## Additional information

### Funding

| Funder | Grant reference number | Author |
| --- | --- | --- |
| Berrow Foundation | Graduate Student Fellowship | Daniel Ballmer |
| Wellcome Trust | https://doi.org/10.35802/210622 | Bungo Akiyoshi |
| Wellcome Trust | 227243/Z/23/Z | Bungo Akiyoshi |
| Wellcome Trust | https://doi.org/10.35802/203149 | Bungo Akiyoshi |

The funders had no role in study design, data collection and interpretation, or the decision to submit the work for publication. For the purpose of Open Access, the authors have applied a CC BY public copyright license to any Author Accepted Manuscript version arising from this submission.

### Author contributions

Daniel Ballmer, Conceptualization, Resources, Formal analysis, Validation, Investigation, Visualization, Methodology, Writing - original draft, Writing – review and editing; Bungo Akiyoshi, Conceptualization, Supervision, Funding acquisition, Investigation, Project administration, Writing – review and editing

### Author ORCIDs

Daniel Ballmer ⓘ http://orcid.org/0000-0002-1966-0960
Bungo Akiyoshi ⓘ https://orcid.org/0000-0001-6010-394X

Reviewer #1 (Public Review): https://doi.org/10.7554/eLife.93522.3.sa1
Reviewer #2 (Public Review): https://doi.org/10.7554/eLife.93522.3.sa2
Reviewer #3 (Public Review): https://doi.org/10.7554/eLife.93522.3.sa3
Author Response https://doi.org/10.7554/eLife.93522.3.sa4

## Additional files

### Supplementary files

• Supplementary file 1. Table containing information (e.g. names, sequences, source, or construction details) on trypanosome cell lines, primers, plasmids, bacmids, and synthetic DNA used in this study.

• Supplementary file 2. Table containing list of proteins (and their iBAQ values) that co-purified with YFP-Aurora B[AUK1], GFP-INCENP[CPC1], GFP-KIN-A, and GFP-KIN-B (*Figure 1*), as well as CLMS results for native chromosomal passenger complexes (CPCs) isolated by immunoprecipitation of YFP-Aurora B[AUK1] (*Figure 3*) and kinetochore proteins isolated by immunoprecipitation of YFP-KKIP1 (KKT8 complex shown in *Figure 4*).

• MDAR checklist

### Data availability

All raw mass spectrometry files and custom database files used in this study have been deposited with the ProteomeXchange Consortium via the PRIDE partner repository (*Deutsch et al., 2020*; *Perez-Riverol et al., 2019*) with the dataset identifier PXD045987.

The following dataset was generated:

| Author(s) | Year | Dataset title | Dataset URL | Database and Identifier |
|---|---|---|---|---|
| Akiyoshi B, Ballmer D | 2023 | Mass spectrometry analysis of chromosomal passenger complex proteins and kinetochore proteins in *Trypanosoma brucei* | https://www.ebi.ac.uk/pride/archive/projects/PXD045987 | PRIDE, PXD045987 |

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

# Appendix 1

## Appendix 1—key resources table

| Reagent type (species) or resource | Designation | Source or reference | Identifiers | Additional information |
|---|---|---|---|---|
| Cell line (*Trypanosoma brucei brucei*) | TREU 927/4 procyclic cells expressing T7 RNA polymerase and the tetracycline repressor to allow inducible expression | *Poon et al., 2012* | SmOxP9 | Background strain used for derivation of cell lines described in *Supplementary file 1* |
| Cell line (*Trypanosoma brucei brucei*) | TREU 927/4 expressing T7 RNA polymerase, tetracycline repressor, and the Cas9 nuclease | *Beneke et al., 2017* | PCF1339 | Background strain used for derivation of cell lines described in *Supplementary file 1* |
| Strain, strain background (*Escherichia coli*) | Rosetta 2(DE3)pLysS | Novagen | 71403 | |
| Strain, strain background (*Escherichia coli*) | BL21(DE3) | Novagen | 69450 | |
| Recombinant DNA reagent | pEnT5-Y, endogenous YFP tagging, hygromycin | *Kelly et al., 2007* | pEnT5-Y (pBA1) | See *Supplementary file 1* for construction details on all plasmids used in this study |
| Recombinant DNA reagent | pBA148, endogenous tdTomato tagging, blasticidin | *Akiyoshi and Gull, 2014* | pBA148 | See *Supplementary file 1* for construction details on all plasmids used in this study |
| Recombinant DNA reagent | pMig96 (pBA152), 256 LacO, integrate at rDNA | *Landeira and Navarro, 2007* | pMig96 (pBA152) | See *Supplementary file 1* for construction details on all plasmids used in this study |
| Recombinant DNA reagent | pJ1339 (1173+Cas9), for CRISPR-Cas9 gene editing | *Beneke et al., 2017* | pJ1339 | See *Supplementary file 1* for construction details on all plasmids used in this study |
| Recombinant DNA reagent | p2T7-177, inducible expression of RNAi constructs (head to head), integrate at 177 bp | *Wickstead et al., 2002* | p2T7-177 (pBA3) | See *Supplementary file 1* for construction details on all plasmids used in this study |
| Recombinant DNA reagent | pRSFDuet-1, expression of one or two target proteins in bacteria | Novagen | pRSFDuet-1, catalog number 71341-3 | See *Supplementary file 1* for construction details on all plasmids used in this study |
| Recombinant DNA reagent | pBA310, inducible expression vector, integrate at 177 bp | *Nerusheva and Akiyoshi, 2016* | pBA310 | See *Supplementary file 1* for construction details on all plasmids used in this study |
| Recombinant DNA reagent | pBA795, inducible GFP-NLS-LacI expression vector, integrate at 177 bp | *Ishii and Akiyoshi, 2020* | pBA795 | See *Supplementary file 1* for construction details on all plasmids used in this study |
| Recombinant DNA reagent | pPOTv7 (pBA1919), (eYFP, Hygromycin) vector for PCR only tagging (POT) of target genes in *Trypanosoma brucei* | *Dean et al., 2015* | pPOTv7 (pBA1919) | See *Supplementary file 1* for construction details on all plasmids used in this study |
| Recombinant DNA reagent | pEnT6-Y (pBA191), endogenous YFP tagging, G418 | Kind gift from Dehua Lai | pEnT6-Y (pBA191) | See *Supplementary file 1* for construction details on all plasmids used in this study |
| Recombinant DNA reagent | pMig75, Tet inducible GFP-LacI, SAT, ClonNAT | Kind gift from Miguel Navarro | pMig75 (pBA150) | See *Supplementary file 1* for construction details on all plasmids used in this study |
| Recombinant DNA reagent | pMig96, rDNA targeting, 256LacO, Phleo | Kind gift from Miguel Navarro | pMig96 (pBA152) | See *Supplementary file 1* for construction details on all plasmids used in this study |

*Appendix 1 Continued on next page*

*Appendix 1 Continued*

| Reagent type (species) or resource | Designation | Source or reference | Identifiers | Additional information |
|---|---|---|---|---|
| Recombinant DNA reagent | BAG164 (shRNA against KIN-A 5' UTR) in pMK-RQ plasmid | Life Technologies Ltd (Invitrogen Division) | NA | See *Supplementary file 1* for sequence information |
| Recombinant DNA reagent | BAG165 (shRNA against KIN-A 3'UTR) in pMK-RQ plasmid | Life Technologies Ltd (Invitrogen Division) | NA | See *Supplementary file 1* for sequence information |
| Recombinant DNA reagent | BAG157 (shRNA against CPC1 5' UTR) in pMK-RQ plasmid | Life Technologies Ltd (Invitrogen Division) | NA | See *Supplementary file 1* for sequence information |
| Recombinant DNA reagent | BAG158 (shRNA against CPC1 3' UTR) in pMK-RQ plasmid | Life Technologies Ltd (Invitrogen Division) | NA | See *Supplementary file 1* for sequence information |
| Recombinant DNA reagent | BAG159 (shRNA against CPC2 5' UTR) in pMK-RQ plasmid | Life Technologies Ltd (Invitrogen Division) | NA | See *Supplementary file 1* for sequence information |
| Recombinant DNA reagent | BAG160 (shRNA against CPC2 3' UTR) in pMK-RQ plasmid | Life Technologies Ltd (Invitrogen Division) | NA | See *Supplementary file 1* for sequence information |
| Recombinant DNA reagent | BAG62 (shRNA against KKT16 CDS) in pMK-RQ plasmid | Life Technologies Ltd (Invitrogen Division) | NA | See *Supplementary file 1* for sequence information |
| Recombinant DNA reagent | BAG80 (shRNA against KKT12 3'UTR) in pMK-RQ plasmid | Life Technologies Ltd (Invitrogen Division) | NA | See *Supplementary file 1* for sequence information |
| Antibody | Mouse monoclonal anti-GFP | Roche | 11814460001 | For immunoprecipitation experiments: 12 µg of antibodies preconjugated with 60 µl slurry of Protein-G magnetic beads |
| Antibody | Rabbit polyclonal anti-GFP | OriGene | TP401 | Dilution for western blot 1:5000 |
| Antibody | Mouse monoclonal TAT1 (anti-trypanosomal-alpha-tubulin) | kind gift from Keith Gull *Woods et al., 1989* | TAT1 | Dilution for western blot 1:5000 |
| Antibody | IRDye 680RD goat anti-mouse | LI-COR | 926-68070 | Dilution for western blot 1:20,000 |
| Antibody | IRDye 800CW goat anti-rabbit | LI-COR | 926-32211 | Dilution for western blot 1:20,000 |
| Commercial assay, kit | Protein-G magnetic beads | Thermo Fisher Scientific | 10004D | |
| Chemical compound, drug | RapiGest | Waters | 186001860 | |
| Peptide, recombinant protein | Trypsin | Promega | V5111 | |
| Chemical compound, drug | BS$^3$ (bis(sulfosuccinimidyl)suberate) | Thermo Fisher Scientific | 21580 | |
| Commercial assay, kit | TALON metal affinity resin | TAKARA BIO EUROPE | 635503 | |
| Commercial assay, kit | Zeba spin desalting columns | Thermo Fisher Scientific | 89883 | |
| Commercial assay, kit | SimplyBlue SafeStain | Life Technologies Ltd (Invitrogen Division) | LC6060 | |
| Peptide, recombinant protein | Porcine tubulin | Cytoskeleton | T-240 | |
| Software, algorithm | TriTrypDB | http://tritrypdb.org/tritrypdb/ | RRID:SCR_007043 | |
| Software, algorithm | MaxQuant (version 2.0.1) | *Cox and Mann, 2008* | RRID:SCR_014485 | |
| Software, algorithm | pLink2 | *Chen et al., 2019* | RRID:SCR_000084 | |
| Software, algorithm | AlphaFold2-Multimer-v2 | *Evans et al., 2022*; *Jumper et al., 2021* | | |

*Appendix 1 Continued on next page*

*Appendix 1 Continued*

| Reagent type (species) or resource | Designation | Source or reference | Identifiers | Additional information |
|---|---|---|---|---|
| Software, algorithm | ColabFold | *Mirdita et al., 2022* | | |
| Software, algorithm | hmmsearch | https://www.ebi.ac.uk/Tools/hmmer/search/hmmsearch | | |
| Software, algorithm | MAFFT | https://mafft.cbrc.jp/alignment/server/index.html | RRID:SCR_011811 | |
| Software, algorithm | Fiji | *Schneider et al., 2012* | RRID:SCR_002285 | |
| Software, algorithm | Python | https://www.python.org/ | RRID:SCR_008394 | |
| Software, algorithm | xiView | *Graham et al., 2019*, https://xiview.org/index.php | | |

