## [Editor Report · eLife assessment]

This **important** study identifies the mitotic localization mechanism for Aurora B and INCENP (parts of the chromosomal passenger complex, CPC) in *Trypanosoma brucei*. The mechanism differs from that in the more commonly studied opisthokonts and is supported by **compelling** RNAi and imaging experiments, targeted mutations, immunoprecipitations with crosslinking/mass spec, and AlphaFold interaction predictions. The findings will be of interest to cell biologists working on cell division, parasitologists, and those interested in the evolution of mitotic mechanisms.

---

## [Referee Report · Reviewer #1 (Public Review)]

Summary:

The CPC plays multiple essential roles in mitosis such as kinetochore-microtubule attachment regulation, kinetochore assembly, spindle assembly checkpoint activation, anaphase spindle stabilization, cytokinesis, and nuclear envelope formation, as it dynamically changes its mitotic localization: it is enriched at inner centromeres from prophase to metaphase but it is relocalized at the spindle midzone in anaphase. The business end of the CPC is Aurora B and its allosteric activation module IN-box, which is located at the C-terminus of INCENP. In most well-studied eukaryotic species, Aurora B activity is locally controlled by the localization module of the CPC, Survivin, Borealin and the N-terminal portion of INCENP. Survivin and Borealin, which bind the N-terminus of INCENP, recognize histone residues that are specifically phosphorylated in mitosis, while anaphase spindle midzone localization is supported by the direct microtubule-binding capacity of the SAH (single alpha helix) domain of INCENP and other microtubule-binding proteins that specifically interact with INCENP during anaphase, which are under the regulation of CDK activity. One of these examples includes the kinesin-like protein MKLP2 in vertebrates. Trypanosoma is an evolutionarily interesting species to study mitosis since its kinetochore and centromere proteins do not show any similarity to other major branches of eukaryotes, while orthologs of Aurora B and INCENP have been identified. Combining molecular genetics, imaging, biochemistry, cross-linking IP-MS (IP-CLMS), and structural modeling, this manuscript reveals that two orphan kinesin-like proteins KIN-A and KIN-B act as localization modules of the CPC in *Trypanosoma brucei*. The IP-CLMS, AlphaFold2 structural predictions, and domain deletion analysis support the idea that (1) KIN-A and KIN-B form a heterodimer via their coiled-coil domains, (2) Two alpha helices of INCENP interact with the coiled-coil of the KIN-A-KIN-B heterodimer, (3) conserved KIN-A C-terminal CD1 and CD2 interact with the heterodimeric KKT9-KKT11 complex, which is a submodule of the KKT7-KKT8 kinetochore complex composed of KKT7, KKT8, KKT9, KKT11, and KKT12 unique to Trypanosoma, (4) KIN-A and KIN-B coiled-coil domains and the KKT7-KKT8 complex are required for CPC localization at the centromere, (5) CD1 and CD2 domains of KIN-A support its centromere localization. The authors further introduced a KIN-A rigor mutant and knocked-down wild-type KIN-A to show that the ATPase activity of KIN-A seems dispensable for centromere targeting but critical for spindle midzone enrichment of the CPC. The imaging data of the KIN-A rigor mutant suggest that dynamic KIN-A-microtubule interaction is required for metaphase alignment of the kinetochores and proliferation. Overall, the study reveals novel pathways of CPC localization regulation via KIN-A and KIN-B by multiple complementary approaches.

Strengths:

The major conclusion is collectively supported by multiple approaches, combining CRISPR-mediated gene deletion and complementation/site specific genome engineering, epistasis analysis of cellular localization, AlphaFold2 structure prediction of protein complexes, IP-CLMS and biochemical reconstitution (the complex of KKT8, KKT9, KKT11 and KKT12)

Weaknesses:

Minor weakness. The authors imply that KIN-A, but not KIN-B, interacts with microtubules based on microtubule pelleting assay (Fig. S6), but the substantial insoluble fractions of 6HIS-KINA and 6HIS-KIN-B make it difficult to conclusively interpret the data. It is possible that these two proteins are not stable unless they form a heterodimer.

---

## [Referee Report · Reviewer #2 (Public Review)]

How the chromosomal passenger complex (CPC) and its subunit Aurora B kinase regulate kinetochore-microtubule attachment, and how the CPC relocates from kinetochores to the spindle midzone as a cell transitions from metaphase to anaphase are questions of great interest. In this study, Ballmer and Akiyoshi take a deep dive into the CPC in *T. brucei*, a kinetoplastid parasite with a kinetochore composition that varies greatly from other organisms.

Using a combination of approaches, most importantly in silico protein predictions using alphafold multimer and light microscopy in dividing *T. brucei*, the authors convincingly present and analyse the composition of the *T. brucei* CPC. This includes the identification of KIN-A and KIN-B, proteins of the kinesin family. This is a clear advancement over earlier work, for example by Li and colleagues in 2008. The involvement of KIN-A and KIN-B is of particular interest, as it provides a clue for the (re)localization of the CPC during the cell cycle. The evolutionary perspective makes the paper potentially interesting for a wide audience of cell biologists, a point that the authors bring across properly in the title, the abstract, and their discussion.

The evolutionary twist of the paper would be strengthened 'experimentally' by predictions of the structure of the CPC beyond *T. brucei*. Depending on how far the authors can extend their in-silico analysis, it would be of interest to discuss (a) available/predicted CPC structures in well-studied organisms and (b) structural predictions in other euglenozoa. What are the general structural properties of the CPC (e.g. flexible linkers, overall dimensions, structural differences when subunits are missing etc.)? How common is the involvement of kinesin-like proteins?

---

## [Referee Report · Reviewer #3 (Public Review)]

Summary:

The protein kinase, Aurora B, is a critical regulator of mitosis and cytokinesis in eukaryotes, exhibiting a dynamic localisation. As part of the Chromosomal Passenger Complex (CPC), along with the Aurora B activator, INCENP, and the CPC localisation module comprised of Borealin and Survivin, Aurora B travels from the kinetochores at metaphase to the spindle midzone at anaphase, which ensures its substrates are phosphorylated in a time- and space-dependent manner. In the kinetoplastid parasite, *T. brucei*, the Aurora B orthologue (AUK1), along with an INCENP orthologue known as CPC1, and a kinetoplastid-specific protein CPC2, also displays a dynamic localisation, moving from the kinetochores at metaphase, to the spindle midzone at anaphase, to the anterior end of the newly synthesised flagellum attachment zone (FAZ) at cytokinesis. However, the trypanosome CPC lacks orthologues of Borealin and Survivin, and *T. brucei* kinetochores also have a unique composition, being comprised of dozens of kinetoplastid-specific proteins (KKTs). Of particular importance for this study are KKT7 and the KKT8 complex (comprising KKT8, KKT9, KKT11, and KKT12). Here, Ballmer and Akiyoshi seek to understand how the CPC assembles and is targeted to its different locations during the cell cycle in *T. brucei*.

Strengths & Weaknesses:

Using immunoprecipitation and mass-spectrometry approaches, Ballmer and Akiyoshi show that AUK1, CPC1, and CPC2 associate with two orphan kinesins, KIN-A and KIN-B, and with the use of endogenously expressed fluorescent fusion proteins, demonstrate for the first time that KIN-A and KIN-B display a dynamic localisation pattern similar to other components of the CPC, providing compelling evidence for KIN-A and KIN-B being bona fide CPC proteins.

They then demonstrate, by using RNAi to deplete individual components, that the CPC proteins have hierarchical interdependencies for their localisation to the kinetochores at metaphase. These experiments appear to have been well performed.

Ballmer and Akiyoshi then go on to determine the kinetochore localisation domains of KIN-A and KIN-B. Using ectopically expressed GFP-tagged truncations, they show that coiled coil domains within KIN-A and KIN-B, as well as a disordered C-terminal tail present only in KIN-A, but not the N-terminal motor domains of KIN-A or KIN-B, are required for kinetochore localisation. These data are strengthened by immunoprecipitating CPC complexes and crosslinking them prior to mass spectrometry analysis (IP-CLMS), a state-of-the-art approach, to determine the contacts between the CPC components. Structural predictions of the CPC structure are also made using AlphaFold2, suggesting that coiled coils form between KIN-A and KIN-B, and that KIN-A/B interact with the N termini of CPC1 and CPC2. Experimental results showing that CPC1 and CPC2 are unable to localise to kinetochores if they lack their N-terminal domains are consistent with these predictions. Altogether these data provide compelling evidence of the protein domains required for CPC kinetochore localisation and CPC protein interactions and indicate that both KIN-A and KIN-B have a role to play.

Next, using a mixture of RNAi depletion and LacI-LacO recruitment experiments, the authors show that kinetochore proteins KKT7 and KKT9 are required for AUK1 to localise to kinetochores (other KKT8 complex components were not tested here) and that all components of the KKT8 complex are required for KIN-A kinetochore localisation. Further, both KKT7 and KKT8 were able to recruit AUK1 to an ectopic locus in S phase, and KKT7 recruited KKT8 complex proteins, indicating it is upstream of KKT8, in line with previous work showing kinetochore localization of KKT7 is unaffected by disruption of the KKT8 complex. This leads to the conclusion that the KKT8 complex is likely the main kinetochore receptor of the CPC.

Further IP-CLMS experiments, in combination with recombinant protein pull down assays and structural predictions, suggested that within the KKT8 complex, there are two subcomplexes of KKT8:KKT12 and KKT9:KKT11, and that KKT7 interacts with KKT9:KKT11 to recruit the remainder of the KKT8 complex. The authors also assess the interdependencies between KKT8 complex components for localisation and expression, showing that all four subunits are required for the assembly of a stable KKT8 complex and present AlphaFold2 structural modelling data to support the two subcomplex model. In general, these data are of high quality and convincing, although it is a shame that data showing the effects of KKT8, KKT9 and KKT12 depletion on KKT11 localisation and abundance could not be presented alongside the reciprocal experiments in Fig S4I-L.

The authors also convincingly show that AlphaFold2 predictions of interactions between KKT9:KKT11 and a conserved domain (CD1) in the C-terminal tail of KIN-A are correct, with CD1 and a second conserved domain, CD2, identified through sequence analysis, acting synergistically to promote KIN-A kinetochore localisation at metaphase, but not being required for KIN-A to move to the central spindle at anaphase. They then hypothesise that the kinesin motor domain of KIN-A (but not KIN-B which is predicted to be inactive based on non-conservation of residues key for activity) determines its central spindle localisation at anaphase through binding to microtubules. In support of this hypothesis, the authors show that KIN-A, but not KIN-B can bind microtubules in vitro and in vivo. However, ectopically expressed GFP-NLS fusions of full length KIN-A or KIN-A motor domain did not localise to the central spindle at anaphase. The authors suggest this is due to the GFP fusion disrupting the ATPase activity of the motor domain, although they provide no evidence that this is the case. Instead, they replace endogenous KIN-A with a predicted ATPase-defective mutant (G210A), showing that while this still localises to kinetochores, the kinetochores were frequently misaligned at metaphase, and that it no longer concentrates at the central spindle (with concomitant mis-localisation of AUK1), causing cells to accumulate at anaphase. From these data, the authors conclude that KIN-A ATPase activity is required for chromosome congression to the metaphase plate and its central spindle localisation at anaphase. While these data are highly suggestive that KIN-A possesses ATPase activity, and that this activity is essential for its function, definitive biochemical evidence of KIN-A's ATPase activity is still lacking.

Impact:

Overall, this work uses a wide range of cutting edge molecular and structural predictive tools to provide a significant amount of new and detailed molecular data that shed light on the composition of the unusual trypanosome CPC and how it is assembled and targeted to different cellular locations during cell division. Given the fundamental nature of this research, it will be of interest to many parasitology researchers as well as cell biologists more generally, especially those working on aspects of mitosis and cell division, and those interested in the evolution of the CPC.

---

## [Author Response]

The following is the authors’ response to the original reviews.

**eLife assessment**
This important study identifies the mitotic localization mechanism for Aurora B and INCENP (parts of the chromosomal passenger complex, CPC) in *Trypanosoma brucei*. The mechanism is different from that in the more commonly studied opisthokonts and there is solid support from RNAi and imaging experiments, targeted mutations, immunoprecipitations with crosslinking/mass spec, and AlphaFold interaction predictions. The results could be strengthened by biochemically testing proposed direct interactions and demonstrating that the targeting protein KIN-A is a motor. The findings will be of interest to parasitology researchers as well as cell biologists working on mitosis and cell division, and those interested in the evolution of the CPC.

We thank the editor and the reviewers for their thorough and positive assessment of our work and the constructive feedback to further improve our manuscript. Please find below our responses to the reviewers’ comments. Please note that the conserved glycine residue in the Switch II helix in KIN-A was mistakenly labelled as G209 in the original manuscript. We now corrected it to G210 in the revised manuscript.

**Public Reviews:**

**Reviewer #1 (Public Review):**
Summary:The CPC plays multiple essential roles in mitosis such as kinetochore-microtubule attachment regulation, kinetochore assembly, spindle assembly checkpoint activation, anaphase spindle stabilization, cytokinesis, and nuclear envelope formation, as it dynamically changes its mitotic localization: it is enriched at inner centromeres from prophase to metaphase but it is relocalized at the spindle midzone in anaphase. The business end of the CPC is Aurora B and its allosteric activation module IN-box, which is located at the C-terminal part of INCENP. In most well-studied eukaryotic species, Aurora B activity is locally controlled by the localization module of the CPC, Survivin, Borealin, and the N-terminal portion of INCENP. Survivin and Borealin, which bind the N terminus of INCENP, recognize histone residues that are specifically phosphorylated in mitosis, while anaphase spindle midzone localization is supported by the direct microtubule-binding capacity of the SAH (single alpha helix) domain of INCENP and other microtubule-binding proteins that specifically interact with INCENP during anaphase, which are under the regulation of CDK activity. One of these examples includes the kinesin-like protein MKLP2 in vertebrates.Trypanosoma is an evolutionarily interesting species to study mitosis since its kinetochore and centromere proteins do not show any similarity to other major branches of eukaryotes, while orthologs of Aurora B and INCENP have been identified. Combining molecular genetics, imaging, biochemistry, cross-linking IP-MS (IP-CLMS), and structural modeling, this manuscript reveals that two orphan kinesin-like proteins KIN-A and KIN-B act as localization modules of the CPC in *Trypanosoma brucei*. The IP-CLMS, AlphaFold2 structural predictions, and domain deletion analysis support the idea that (1) KIN-A and KIN-B form a heterodimer via their coiled-coil domain, (2) Two alpha helices of INCENP interact with the coiled-coil of the KIN-A-KIN-B heterodimer, (3) the conserved KIN-A C-terminal CD1 interacts with the heterodimeric KKT9-KKT11 complex, which is a submodule of the KKT7-KKT8 kinetochore complex unique to Trypanosoma, (4) KIN-A and KIN-B coiled-coil domains and the KKT7-KKT8 complex are required for CPC localization at the centromere, (5) CD1 and CD2 domains of KIN-A support its centromere localization. The authors further show that the ATPase activity of KIN-A is critical for spindle midzone enrichment of the CPC. The imaging data of the KIN-A rigor mutant suggest that dynamic KIN-A-microtubule interaction is required for metaphase alignment of the kinetochores and proliferation. Overall, the study reveals novel pathways of CPC localization regulation via KIN-A and KIN-B by multiple complementary approaches.Strengths:The major conclusion is collectively supported by multiple approaches, combining site-specific genome engineering, epistasis analysis of cellular localization, AlphaFold2 structure prediction of protein complexes, IP-CLMS, and biochemical reconstitution (the complex of KKT8, KKT9, KKT11, and KKT12).

We thank the reviewer for her/his positive assessment of our manuscript.

Weaknesses:The predictions of direct interactions (e.g. INCENP with KIN-A/KIN-B, or KIN-A with KKT9-KKT11) have not yet been confirmed experimentally, e.g. by domain mutagenesis and interaction studies.

Thank you for this point. It is true that we do not have evidence for direct interactions between KIN-A with KKT9-KKT11. However, the interaction between INCENP with KIN-A/KIN-B is strongly supported by our cross-linking IP-MS of native complexes. Furthermore, we show that deletion of the INCENPCPC1 N-terminus predicted to interact with KIN-A:KIN-B abolishes kinetochore localization.

The criteria used to judge a failure of localization are not clearly explained (e.g., Figure 5F, G).

As suggested by the reviewer in recommendation #14, we have now included example images for each category (‘kinetochores’, ‘kinetochores + spindle’, ‘spindle’) along with a schematic illustration in Fig. 5F.

It remains to be shown that KIN-A has motor activity.

We thank the reviewer for this important comment. Indeed, motor activity remains to demonstrated using an in vitro system, which is beyond the scope of this study. What we show here is that the motor domain of KIN-A effectively co-sediments with microtubules and that spindle localization of KIN-A is abolished upon deletion of the motor domain. Moreover, mutation of a conserved Glycine residue in the Switch II region (G210) to Alanine (‘rigor mutation’, (Rice et al., 1999)), renders KIN-A incapable of translocating to the central spindle, suggesting that its ATPase activity is required for this process. To clarify this point in the manuscript, we have replaced all instances, where we refer to ‘motor activity’ of KIN-A with ‘ATPase activity’ when referring to experiments performed using the KIN-A rigor mutant. In addition, we have included a Multiple Sequence Alignment (MSA) of KIN-A and KIN-B from different kinetoplastids with human Kinesin-1, human Mklp2 and yeast Klp9 in Figure 6A and S6A, showing the conservation of key motifs required for ATP coordination and tubulin interaction. In the corresponding paragraph in the main text, we describe these data as follows:

‘We therefore speculated that anaphase translocation of the kinetoplastid CPC to the central spindle may involve the kinesin motor domain of KIN-A. KIN-B is unlikely to be a functional kinesin based on the absence of several well-conserved residues and motifs within the motor domain, which are fully present in KIN-A (Li et al., 2008). These include the P-loop, switch I and switch II motifs, which form the nucleotide binding cleft, and many conserved residues within the α4-L12 elements, which interact with tubulin (Fig. S6A) (Endow et al., 2010). Consistent with this, the motor domain of KIN-B, contrary to KIN-A, failed to localize to the mitotic spindle when expressed ectopically (Fig. S2E) and did not co-sediment with microtubules in our in vitro assay (Fig. S6B).’

The authors imply that KIN-A, but not KIN-B, interacts with microtubules based on microtubule pelleting assay (Fig. S6), but the substantial insoluble fractions of 6HIS-KINA and 6HIS-KIN-B make it difficult to conclusively interpret the data. It is possible that these two proteins are not stable unless they form a heterodimer.

This is indeed a possibility. We are currently aiming at purifying full-length recombinant KIN-A and KIN-B (along with the other CPC components), which will allow us to perform in vitro interaction studies and to investigate biochemical properties of this complex (including the role of the motor domains of KIN-A and KIN-B) within the framework of an in-depth follow-up study. To address the point above, we have added the following text in the legend corresponding to Fig. S6:

‘Microtubule co-sedimentation assay with 6HIS-KIN-A2-309 (left) and 6HIS-KIN-B2-316 (right). S and P correspond to supernatant and pellet fractions, respectively. Note that both constructs to some extent sedimented even in the absence of microtubules. Hence, lack of microtubule binding for KIN-B may be due to the unstable non-functional protein used in this study.’

For broader context, some prior findings should be introduced, e.g. on the importance of the microtubule-binding capacity of the INCENP SAH domain and its regulation by mitotic phosphorylation (PMID 8408220, 26175154, 26166576, 28314740, 28314741, 21727193), since KIN-A and KIN-B may substitute for the function of the SAH domain.

We have modified the introduction to include the following text and references mentioned by the reviewer:‘The localization module comprises Borealin, Survivin and the N-terminus of INCENP, which are connected to one another via a three-helical bundle (Jeyaprakash et al., 2007, 2011; Klein et al., 2006). The two modules are linked by the central region of INCENP, composed of an intrinsically disordered domain and a single alpha helical (SAH) domain. INCENP harbours microtubule-binding domains within the N-terminus and the central SAH domain, which play key roles for CPC localization and function (Samejima et al., 2015; Kang et al., 2001; Noujaim et al., 2014; Cormier et al., 2013; Wheatley et al., 2001; Nakajima et al., 2011; Fink et al., 2017; Wheelock et al., 2017; van der Horst et al., 2015; Mackay et al., 1993).’

**Reviewer #2 (Public Review):**
How the chromosomal passenger complex (CPC) and its subunit Aurora B kinase regulate kinetochore-microtubule attachment, and how the CPC relocates from kinetochores to the spindle midzone as a cell transitions from metaphase to anaphase are questions of great interest. In this study, Ballmer and Akiyoshi take a deep dive into the CPC in *T. brucei*, a kinetoplastid parasite with a kinetochore composition that varies greatly from other organisms.Using a combination of approaches, most importantly in silico protein predictions using alphafold multimer and light microscopy in dividing *T. brucei*, the authors convincingly present and analyse the composition of the *T. brucei* CPC. This includes the identification of KIN-A and KIN-B, proteins of the kinesin family, as targeting subunits of the CPC. This is a clear advancement over earlier work, for example by Li and colleagues in 2008. The involvement of KIN-A and KIN-B is of particular interest, as it provides a clue for the (re)localization of the CPC during the cell cycle. The evolutionary perspective makes the paper potentially interesting for a wide audience of cell biologists, a point that the authors bring across properly in the title, the abstract, and their discussion.The evolutionary twist of the paper would be strengthened 'experimentally' by predictions of the structure of the CPC beyond *T. brucei*. Depending on how far the authors can extend their in-silico analysis, it would be of interest to discuss (a) available/predicted CPC structures in well-studied organisms and (b) structural predictions in other euglenozoa. What are the general structural properties of the CPC (e.g. flexible linkers, overall dimensions, structural differences when subunits are missing etc.)? How common is the involvement of kinesin-like proteins? In line with this, it would be good to display the figure currently shown as S1D (or similar) as a main panel.

We thank the reviewer for her/his encouraging assessment of our manuscript and the appreciation on the extent of the evolutionary relevance of our work. As suggested, we have moved the phylogenetic tree previously shown in Fig. S1D to the main Fig. 1F. Our AF2 analysis of CPC proteins and (sub)complexes from other kinetoplastids failed to predict reliable interactions among CPC proteins except for that between Aurora B and the IN box. It therefore remains unclear whether CPC structures are conserved among kinetoplastids. Because components of CPC remain unknown in other euglenozoa (other than Aurora B and INCENP), we cannot perform structural predictions of CPC in diplonemids or euglenids.

It remains unclear how common the involvement of kinesin-like proteins with the CPC is in other eukaryotes, partly because we could not identify an obvious homolog of KIN-A/KIN-B outside of kinetoplastids. Addressing this question would require experimental approaches in various eukaryotes (e.g. immunoprecipitation and mass spectrometry of Aurora B) as we carried out in this manuscript using *Trypanosoma brucei*.

**Reviewer #3 (Public Review):**
Summary:The protein kinase, Aurora B, is a critical regulator of mitosis and cytokinesis in eukaryotes, exhibiting a dynamic localisation. As part of the Chromosomal Passenger Complex (CPC), along with the Aurora B activator, INCENP, and the CPC localisation module comprised of Borealin and Survivin, Aurora B travels from the kinetochores at metaphase to the spindle midzone at anaphase, which ensures its substrates are phosphorylated in a time- and space-dependent manner. In the kinetoplastid parasite, *T. brucei*, the Aurora B orthologue (AUK1), along with an INCENP orthologue known as CPC1, and a kinetoplastid-specific protein CPC2, also displays a dynamic localisation, moving from the kinetochores at metaphase to the spindle midzone at anaphase, to the anterior end of the newly synthesised flagellum attachment zone (FAZ) at cytokinesis. However, the trypanosome CPC lacks orthologues of Borealin and Survivin, and *T. brucei* kinetochores also have a unique composition, being comprised of dozens of kinetoplastid-specific proteins (KKTs). Of particular importance for this study are KKT7 and the KKT8 complex (comprising KKT8, KKT9, KKT11, and KKT12). Here, Ballmer and Akiyoshi seek to understand how the CPC assembles and is targeted to its different locations during the cell cycle in *T. brucei*.Strengths & Weaknesses:Using immunoprecipitation and mass-spectrometry approaches, Ballmer and Akiyoshi show that AUK1, CPC1, and CPC2 associate with two orphan kinesins, KIN-A and KIN-B, and with the use of endogenously expressed fluorescent fusion proteins, demonstrate for the first time that KIN-A and KIN-B display a dynamic localisation pattern similar to other components of the CPC. Most of these data provide convincing evidence for KIN-A and KIN-B being bona fide CPC proteins, although the evidence that KIN-A and KIN-B translocate to the anterior end of the new FAZ at cytokinesis is weak - the KIN-A/B signals are very faint and difficult to see, and cell outlines/brightfield images are not presented to allow the reader to determine the cellular location of these faint signals (Fig S1B).

We thank the reviewer for their thorough assessment of our manuscript and the insightful feedback to further improve our study. To address the point above, we have acquired new microscopy data for Fig. S1B and S1C, which now includes phase contrast images, and have chosen representative cells in late anaphase and telophase. We hope that the signal of Aurora BAUK1, KIN-A and KIN-B at the anterior end of the new FAZ can be now distinguished more clearly.

They then demonstrate, by using RNAi to deplete individual components, that the CPC proteins have hierarchical interdependencies for their localisation to the kinetochores at metaphase. These experiments appear to have been well performed, although only images of cell nuclei were shown (Fig 2A), meaning that the reader cannot properly assess whether CPC components have localised elsewhere in the cell, or if their abundance changes in response to depletion of another CPC protein.

We chose to show close-ups of the nucleus to highlight the different localization patterns of CPC proteins under the different RNAi conditions. In none of these conditions did we observe mis-localization of CPC subunits to the cytoplasm. To clarify this point, we added the following sentence in the legend for Figure 2A:

‘(A) Representative fluorescence micrographs showing the localization of YFP-tagged Aurora BAUK1, INCENPCPC1, KIN-A and KIN-B in 2K1N cells upon RNAi-mediated knockdown of indicated CPC subunits. Note that nuclear close-ups are shown here. CPC proteins were not detected in the cytoplasm. RNAi was induced with 1 μg/mL doxycycline for 24 h (KIN-B RNAi) or 16 h (all others). Cell lines: BAP3092, BAP2552, BAP2557, BAP3093, BAP2906, BAP2900, BAP2904, BAP3094, BAP2899, BAP2893, BAP2897, BAP3095, BAP3096, BAP2560, BAP2564, BAP3097. Scale bars, 2 μm.’

Ballmer and Akiyoshi then go on to determine the kinetochore localisation domains of KIN-A and KIN-B. Using ectopically expressed GFP-tagged truncations, they show that coiled-coil domains within KIN-A and KIN-B, as well as a disordered C-terminal tail present only in KIN-A, but not the N-terminal motor domains of KIN-A or KIN-B, are required for kinetochore localisation. These data are strengthened by immunoprecipitating CPC complexes and crosslinking them prior to mass spectrometry analysis (IP-CLMS), a state-of-the-art approach, to determine the contacts between the CPC components. Structural predictions of the CPC structure are also made using AlphaFold2, suggesting that coiled coils form between KIN-A and KIN-B, and that KIN-A/B interact with the N termini of CPC1 and CPC2. Experimental results show that CPC1 and CPC2 are unable to localise to kinetochores if they lack their N-terminal domains consistent with these predictions. Altogether these data provide convincing evidence of the protein domains required for CPC kinetochore localisation and CPC protein interactions. However, the authors also conclude that KIN-B plays a minor role in localising the CPC to kinetochores compared to KIN-A. This conclusion is not particularly compelling as it stems from the observation that ectopically expressed GFP-NLS-KIN-A (full length or coiled-coil domain + tail) is also present at kinetochores during anaphase unlike endogenously expressed YFP-KIN-A. Not only is this localisation probably an artifact of the ectopic expression, but the KIN-B coiled-coil domain localises to kinetochores from S to metaphase and Fig S2G appears to show a portion of the expressed KIN-B coiled-coil domain colocalising with KKT2 at anaphase. It is unclear why KIN-B has been discounted here.

As the reviewer points out, a small fraction of GFP-NLS-KIN-B317-624 is indeed detectable at kinetochores in anaphase, although most of the protein shows diffuse nuclear staining. There are various explanations for this phenomenon: It is conceivable that the KIN-B motor domain may contribute to microtubule binding and translocation of the CPC from kinetochores onto the spindle in anaphase. In our experiments, ectopically expressed KIN-B317-624 likely outcompetes a fraction of endogenous KIN-B for binding to KIN-A, which could interfere with this translocation process, leaving a population of CPC ‘stranded’ at kinetochores in anaphase. Another possibility, hinted at by the reviewer, is that the C-terminus of KIN-B interacts with receptors at the kinetochore/centromere. Although we do not discount this possibility, we nevertheless decided to focus on KIN-A in this study, because the anaphase kinetochore retention phenotype for both full-length GFP-NLS-KIN-A and -KIN-A309-862 is much stronger than for KIN-B317-624. Two additional reasons were that (i) KIN-A is highly conserved within kinetoplastids, whereas KIN-B orthologs are missing in some kinetoplastids, and (ii) no convincing interactions between KIN-B and kinetochore proteins were predicted by AF2.

To address the reviewer’s point, we decided to include KIN-B in the title of this manuscript, which now reads: ‘Dynamic localization of the chromosomal passenger complex is controlled by the orphan kinesins KIN-A and KIN-B in the kinetoplastid parasite *Trypanosoma brucei*’.

Moreover, we modified the corresponding paragraph in the results section as follows:

‘Intriguingly, unlike endogenously YFP-tagged KIN-A, ectopically expressed GFP fusions of both full-length KIN-A and KIN-A310-862 clearly localized at kinetochores even in anaphase (Figs. 2, F and H). Weak anaphase kinetochore signal was also detectable for KIN-B317-624 (Fig. S2F). GFP fusions of the central coiled-coil domain or the C-terminal disordered tail of KIN-A did not localize to kinetochores (data not shown). These results show that kinetochore localization of the CPC is mediated by KIN-A and KIN-B and requires both the central coiled-coil domain as well as the C-terminal disordered tail of KIN-A.’

Next, using a mixture of RNAi depletion and LacI-LacO recruitment experiments, the authors show that kinetochore proteins KKT7 and KKT9 are required for AUK1 to localise to kinetochores (other KKT8 complex components were not tested here) and that all components of the KKT8 complex are required for KIN-A kinetochore localisation. Further, both KKT7 and KKT8 were able to recruit AUK1 to an ectopic locus in the S phase, and KKT7 recruited KKT8 complex proteins, which the authors suggest indicates it is upstream of KKT8. However, while these experiments have been performed well, the reciprocal experiment to show that KKT8 complex proteins cannot recruit KKT7, which could have confirmed this hierarchy, does not appear to have been performed. Further, since the LacI fusion proteins used in these experiments were ectopically expressed, they were retained (artificially) at kinetochores into anaphase; KKT8 and KIN-A were both able to recruit AUK1 to LacO foci in anaphase, while KKT7 was not. The authors conclude that this suggests the KKT8 complex is the main kinetochore receptor of the CPC - while very plausible, this conclusion is based on a likely artifact of ectopic expression, and for that reason, should be interpreted with a degree of caution.

We previously showed that RNAi-mediated depletion of KKT7 disrupts kinetochore localization of KKT8 complex members, whereas kinetochore localization of KKT7 is unaffected by disruption of the KKT8 complex (Ishii and Akiyoshi, 2020). Moreover, in contrast to the KKT8 complex, KKT7 remains at kinetochores in anaphase (Akiyoshi and Gull, 2014). These data show that KKT7 is upstream of the KKT8 complex. In this context, the LacI-LacO tethering approach can be very useful to probe whether two proteins (or domains of proteins) could interact in vivo either directly or indirectly. However, a recruitment hierarchy cannot be inferred from such experiments because the data just shows whether X can recruit Y to an ectopic locus (but not whether X is upstream of Y or vice versa). Regarding the retention of Aurora BAUK1 at kinetochores in anaphase upon ectopic expression of GFP-KKT8-LacI, we agree with the reviewer that these data need to be carefully interpreted. Nevertheless, the notion that the KKT7-KKT8 complex recruits the CPC to kinetochores is also strongly supported by IP-MS, RNAi experiments, and AF2 predictions. For clarification and to address the reviewer’s point, we re-formulated the corresponding paragraph in the main text:

‘We previously showed that KKT7 lies upstream of the KKT8 complex (Ishii and Akiyoshi, 2020). Indeed, GFP-KKT72-261-LacI recruited tdTomato-KKT8, -KKT9 and -KKT12 (Fig. S4E). Expression of both GFP-KKT72-261-LacI and GFP-KKT8-LacI resulted in robust recruitment of tdTomato-Aurora BAUK1 to LacO foci in S phase (Figs. 4, E and F). Intriguingly, we also noticed that, unlike endogenous KKT8 (which is not present in anaphase), ectopically expressed GFP-KKT8-LacI remained at kinetochores during anaphase (Fig. 4F). This resulted in a fraction of tdTomato-Aurora BAUK1 being trapped at kinetochores during anaphase instead of migrating to the central spindle (Fig. 4F). We observed a comparable situation upon ectopic expression of GFP-KIN-A, which is retained on anaphase kinetochores together with tdTomato-KKT8 (Fig. S4F). In contrast, Aurora BAUK1 was not recruited to LacO foci marked by GFP- KKT72-261-LacI in anaphase (Fig. 4E).’

Further IP-CLMS experiments, in combination with recombinant protein pull-down assays and structural predictions, suggested that within the KKT8 complex, there are two subcomplexes of KKT8:KKT12 and KKT9:KKT11, and that KKT7 interacts with KKT9:KKT11 to recruit the remainder of the KKT8 complex. The authors also assess the interdependencies between KKT8 complex components for localisation and expression, showing that all four subunits are required for the assembly of a stable KKT8 complex and present AlphaFold2 structural modelling data to support the two subcomplex models. In general, these data are of high quality and convincing with a few exceptions. The recombinant pulldown assay (Fig. 4H) is not particularly convincing as the 3rd eluate gel appears to show a band at the size of KKT11 (despite the labelling indicating no KKT11 was present in the input) but no pulldown of KKT9, which was present in the input according to the figure legend (although this may be mislabeled since not consistent with the text). The text also states that 6HIS-KKT8 was insoluble in the absence of KKT12, but this is not possible to assess from the data presented.

We thank the reviewer for pointing out an error in the text: ‘Removal of both KKT9 and KKT11 did not impact formation of the KKT8:KKT12 subcomplex’ should read ‘Removal of either KKT9 or KKT11 did not impact formation of the KKT8:KKT12 subcomplex’. Regarding the very faint band perceived to be KKT11 in the 3rd eluate: This band runs slightly lower than KKT11 and likely represents a bacterial contaminant (which we have seen also in other preps in the past). We have made a note of this in the corresponding legend (new Fig. 4I). Moreover, we provide the estimated molecular weights for each subunit, as suggested by the reviewer in recommendation #14 (see below):

‘(I) Indicated combinations of 6HIS-tagged KKT8 (~46 kDa), KKT9 (~39 kDa), KKT11 (~29 kDa) and KKT12 (~23 kDa) were co-expressed in *E. coli*, followed by metal affinity chromatography and SDS-PAGE. The asterisk indicates a common contaminant.’

The corresponding paragraph in the results section now reads:

To validate these findings, we co-expressed combinations of 6HIS-KKT8, KKT9, KKT11 and KKT12 in *E. coli* and performed metal affinity chromatography (Fig. 4I). 6HIS-KKT8 efficiently pulled down KKT9, KKT11 and KKT12, as shown previously (Ishii and Akiyoshi, 2020). In the absence of KKT9, 6HIS-KKT8 still pulled down KKT11 and KKT12. Removal of either KKT9 or KKT11 did not impact formation of the KKT8:KKT12 subcomplex. In contrast, 6HIS-KKT8 could not be recovered without KKT12, indicating that KKT12 is required for formation of the full KKT8 complex. These results support the idea that the KKT8 complex consists of KKT8:KKT12 and KKT9:KKT11 subcomplexes.’

It is also surprising that data showing the effects of KKT8, KKT9, and KKT12 depletion on KKT11 localisation and abundance are not presented alongside the reciprocal experiments in Fig S4G-J.

YFP-KKT11 is delocalized upon depletion of KKT8 and KKT9 (see below). Unfortunately, we were unsuccessful in our attempts at deriving the corresponding KKT12 RNAi cell line, rendering this set of data incomplete. Because these data are not of critical importance for this study, we decided not to invest more time in attempting further transfections.

**Author response image 1. sa4fig1:** 

The authors also convincingly show that AlphaFold2 predictions of interactions between KKT9:KKT11 and a conserved domain (CD1) in the C-terminal tail of KIN-A are likely correct, with CD1 and a second conserved domain, CD2, identified through sequence analysis, acting synergistically to promote KIN-A kinetochore localisation at metaphase, but not being required for KIN-A to move to the central spindle at anaphase. They then hypothesise that the kinesin motor domain of KIN-A (but not KIN-B which is predicted to be inactive based on non-conservation of residues key for activity) determines its central spindle localisation at anaphase through binding to microtubules. In support of this hypothesis, the authors show that KIN-A, but not KIN-B can bind microtubules in vitro and in vivo. However, ectopically expressed GFP-NLS fusions of full-length KIN-A or KIN-A motor domain did not localise to the central spindle at anaphase. The authors suggest this is due to the GPF fusion disrupting the ATPase activity of the motor domain, but they provide no evidence that this is the case. Instead, they replace endogenous KIN-A with a predicted ATPase-defective mutant (G209A), showing that while this still localises to kinetochores, the kinetochores were frequently misaligned at metaphase, and that it no longer concentrates at the central spindle (with concomitant mis-localisation of AUK1), causing cells to accumulate at anaphase. From these data, the authors conclude that KIN-A ATPase activity is required for chromosome congression to the metaphase plate and its central spindle localisation at anaphase. While potentially very interesting, these data are incomplete in the absence of any experimental data to show that KIN-A possesses ATPase activity or that this activity is abrogated by the G209A mutation, and the conclusions of this section are rather speculative.

Thank you for this important comment, which relates to a similar point raised by Reviewer 1 (see above). Indeed, ATPase and motor activity of KIN-A remain to demonstrated biochemically using recombinant proteins, which is beyond the scope of this study. We generated MSAs of KIN-A and KIN-B from different kinetoplastids with human Kinesin-1, human Mklp2 and yeast Klp9, which are now presented in Figure 6A and S6A. These clearly show that key motifs required for ATP or tubulin binding in other kinesins are highly conserved in KIN-A (but not KIN-B). This includes the conserved glycine residue in the Switch II helix (G234 in human Kinesin-1, G210 in *T. brucei* KIN-A), which forms a hydrogen bond with the γ-phosphate of ATP, and upon mutation has been shown to impair ATPase activity and trap the motor head in a strong microtubule (‘rigor’) state (Rice et al., 1999; Sablin et al., 1996). The prominent rigor phenotype of KIN-AG210A is consistent with KIN-A having ATPase activity. In addition to the data in Fig. 6A and S6A, we made following changes to the main text:

‘We therefore speculated that anaphase translocation of the kinetoplastid CPC to the central spindle may involve the kinesin motor domain of KIN-A. KIN-B is unlikely to be a functional kinesin based on the absence of several well-conserved residues and motifs within the motor domain, which are fully present in KIN-A (Li et al., 2008). These include the P-loop, switch I and switch II motifs, which form the nucleotide binding cleft, and many conserved residues within the α4-L12 elements, which interact with tubulin (Fig. S6A) (Endow et al., 2010). Consistent with this, the motor domain of KIN-B, contrary to KIN-A, failed to localize to the mitotic spindle when expressed ectopically (Fig. S2E) and did not co-sediment with microtubules in our in vitro assay (Fig. S6B).

Ectopically expressed GFP-KIN-A and -KIN-A2-309 partially localized to the mitotic spindle but failed to concentrate at the midzone during anaphase (Figs. 2, F and G), suggesting that N-terminal tagging of the KIN-A motor domain may interfere with its function. To address whether the ATPase activity of KIN-A is required for central spindle localization of the CPC, we replaced one allele of KIN-A with a C-terminally YFP-tagged G210A ATP hydrolysis-defective rigor mutant (Fig. 6A) (Rice et al., 1999) and used an RNAi construct directed against the 3’UTR of KIN-A to deplete the untagged allele. The rigor mutation did not affect recruitment of KIN-A to kinetochores (Figs. S6, C and D). However, KIN-AG210A-YFP marked kinetochores were misaligned in ~50% of cells arrested in metaphase, suggesting that ATPase activity of KIN-A promotes chromosome congression to the metaphase plate (Figs. S6, E-H).’

Impact:Overall, this work uses a wide range of cutting-edge molecular and structural predictive tools to provide a significant amount of new and detailed molecular data that shed light on the composition of the unusual trypanosome CPC and how it is assembled and targeted to different cellular locations during cell division. Given the fundamental nature of this research, it will be of interest to many parasitology researchers as well as cell biologists more generally, especially those working on aspects of mitosis and cell division, and those interested in the evolution of the CPC.

We thank the reviewer for his/her feedback and thoughtful and thorough assessment of our study.

**Recommendations for the authors:**

**Reviewer #1 (Recommendations For The Authors):**
(1) Why did the authors omit KIN-B from the title?

We decided to add KIN-B in the title. Please see our response to Reviewer #3 (public review).

(2) Abstract, line 28, "Furthermore, the kinesin motor activity of KIN-A promotes chromosome alignment in prometaphase and CPC translocation to the central spindle upon anaphase onset." This must be revised - see public review.

We changed this section of the abstract as follows:

‘Furthermore, the ATPase activity of KIN-A promotes chromosome alignment in prometaphase and CPC translocation to the central spindle upon anaphase onset. Thus, KIN-A constitutes a unique ‘two-in-one’ CPC localization module in complex with KIN-B, which directs the CPC to kinetochores (from S phase until metaphase) via its C-terminal tail, and to the central spindle (in anaphase) via its N-terminal kinesin motor domain.’

(3) Line 87-90. The findings by Li et al., 2008 (KIN-A and KIN-B interacting with Aurora B and epistasis analysis) should be introduced more comprehensively in the Introduction section.

We added the following sentence in the introduction:

‘In addition, two orphan kinesins, KIN-A and KIN-B, have been proposed to transiently associate with Aurora BAUK1 during mitosis (Li et al., 2008; Li, 2012).’

(4) Figure 1B. The way the Trypanosoma cell cycle is defined should be briefly explained in the main text, rather than just referring to the figure.

The ‘KN’ annotation of the trypanosome cell cycle is explained in the Figure 1 legend. We now also added a brief description in the main text:

‘We next assessed the localization dynamics of fluorescently tagged KIN-A and KIN-B over the course of the cell cycle (Figs. 1, B-E). *T. brucei* possesses two DNA-containing organelles, the nucleus (‘N’) and the kinetoplast (‘K’). The kinetoplast is an organelle found uniquely in kinetoplastids, which contains the mitochondrial DNA and replicates and segregates prior to nuclear division. The ‘KN’ configuration serves as a good cell cycle marker (Woodward and Gull, 1990; Siegel et al., 2008).’

(5) Line 118. Throughout the paper, it is not clear why GFP-NLS fusion was used instead of GFP fusion. Please justify the fusion of NLS.

NLS refers to a short ‘nuclear localization signal’ (TGRGHKRSREQ) (Marchetti et al., 2000), which ensures that the ectopically expressed construct is imported into the nucleus. When we previously expressed truncations of KKT2 and KKT3 kinetochore proteins, many fragments did not go into the nucleus presumably due to the lack of an NLS, which prevented us from determining which domains are responsible for their kinetochore localization. We have since then consistently used this short NLS sequence in our inducible GFP fusions in the past without any complications. We added a sentence in the Materials & Methods section under Trypanosome culture: ‘All constructs for ectopic expression of GFP fusion proteins include a short nuclear localization signal (NLS) (Marchetti et al., 2000).’ To avoid unnecessary confusion, we removed ‘NLS’ from the main text and figures.

(6) Line 121, "Unexpectedly". It is not clear why this was unexpected.

To clarify this point, we modified this paragraph in the results section:

‘To our surprise, KIN-A-YFP and GFP-KIN-B exhibited a CPC-like localization pattern identical to that of Aurora BAUK1: Both kinesins localized to kinetochores from S phase to metaphase, and then translocated to the central spindle in anaphase (Figs. 1, C-E). Moreover, like Aurora BAUK1, a population of KIN-A and KIN-B localized at the new FAZ tip from late anaphase onwards (Figs. S1, B and C). This was unexpected, because KIN-A and KIN-B were previously reported to localize to the spindle but not to kinetochores or the new FAZ tip (Li et al., 2008). These data suggest that KIN-A and KIN-B are bona fide CPC proteins in trypanosomes, associating with AuroraAUK1, INCENPCPC1 and CPC2 throughout the cell cycle.’

(7) Line 127-129. Defining homologs and orthologs is tricky - there are many homologs and paralogs of kinesin-like proteins. The method to define the presence or absence of KIN-A/KIN-B homologs should be described in the Materials and Methods section.

Due to the difficulty in defining true orthologs for kinesin-like proteins, we took a conservative approach: reciprocal best BLAST hits. We first searched KIN-A homologs using BLAST in the TriTryp database or using hmmsearch using manually prepared hmm profiles. When the top hit in a given organism found *T. brucei* KIN-A in a reciprocal BLAST search in *T. brucei* proteome, we considered the hit as a true ortholog. We modified the Materials and Methods section as below.

‘Searches for homologous proteins were done using BLAST in the TriTryp database (Aslett et al., 2010) or using hmmsearch using manually prepared hmm profiles (HMMER version 3.0; Eddy, 1998). The top hit was considered as a true ortholog only if the reciprocal BLAST search returned the query protein in *T. brucei*.’

(8) Line 156. For non-experts of Trypanosoma cell biology, it is not clear how the nucleolar localization is defined.

The nucleolus in *T. brucei* is discernible as a DAPI-dim region in the nucleus.

(9) Fig.2G and Fig.S2F. These data imply that the coiled-coil and C-terminal tail domains of KIN-A/KIN-B are important for anaphase spindle midzone enrichment. However, it is odd that this was not mentioned. This reviewer recommends that the authors quantify the midzone localization data of these constructs and discuss the role of the coiled-coil domains.

One possibility is that KIN-A and KIN-B need to form a complex (via their coiled-coil domains) to localize to the spindle midzone. Another likely possibility, which is discussed in the manuscript, is that N-terminal tagging of KIN-A impairs motor activity. This is supported by the fact that the central spindle localization is also disrupted in full-length GFP-KIN-A. We decided not to provide a quantification for these data due to low sample sizes for some of the constructs (e.g. expression not observed in all cells).

(10) Line 288-289, "pLDDT scores improved significantly for KIN-A CD1 in complex with KKT9:KKT11 (>80) compared to KIN-A CD1 alone (~20) (Figs. S3, A and B)." I can see that pLDDT score is about 20 at KIN-A CD1 from Figs S3A, but the basis of pLDDT > 80 upon inclusion go KKT9:KKT11 is missing.

We added the pLDDT and PAE plots for the AF2 prediction of KIN-A700-800 in complex with KKT9:KKT11 in Fig. S5B.

(11) Fig. 5A. Since there is no supporting biochemical data for KIN-A-KKT9-KKT11 interaction, it is important to assess the stability of AlphaFold-based structural predictions of the KIN-A-KKT9-KKT11 interaction. Are there significant differences among the top 5 prediction results, and do these interactions remain stable after the "simulated annealing" process used in the AlphaFold predictions? Are predicted CD1-interacting regions/amino residues in KKT9 and KKT11 evolutionarily conserved?

See above. The interaction was predicted in all 5 predictions as shown in Fig. S5B.Conservation of the CD1-interacting regions in KKT9 and KKT11 are shown below:

**Author response image 2. sa4fig2:** KKT9 (residues ~53 – 80 predicted to interact with KIN-A in *T. brucei*).

**Author response image 3. sa4fig3:** KKT11 (residues 61-85 predicted to interact with KIN-A in *T. brucei*).

(12) Line 300, Fig. S5D and E, "failed to localize at kinetochores". From this resolution of the microscopy images, it is not clear if these proteins fail to localize at kinetochores as the KKT and KIN-A310-716 signals overlap. Perhaps, "failed to enrich at kinetochores" is a more appropriate statement.

We changed this sentence according to the reviewer’s suggestion.

(13) Line 309 and Fig 5D and F, "predominantly localized to the mitotic spindle". From this image shown in Fig 5D, it is not clear if KIN-A∆CD1-YFP and Aurora B are predominantly localized to the spindle or if they are still localized to centromeres that are misaligned on the spindle. Without microtubule staining, it is also not clear how microtubules are distributed in these cells. Please clarify how the presence or absence of kinetochore/spindle localization was defined.

As shown in Fig. S5E and S5F, deletion of CD1 clearly impairs kinetochore localization of KIN-A (kinetochores marked by tdTomato-KKT2). Moreover, misalignment of kinetochores, as observed upon expression of the KIN-AG210A rigor mutant, would result in an increase in 2K1N cells and proliferation defects, which is not the case for the KIN-A∆CD1 mutant (Fig. 5H, Fig. S5I). KIN-A∆CD1-YFP appears to localize diffusely along the entire length of the mitotic spindle, whereas we still observe kinetochore-like foci in the rigor mutant. Unfortunately, we do not have suitable antibodies that would allow us to distinguish spindle microtubules from the vast subpellicular microtubule array present in *T. brucei* and hence need to rely on tagging spindle-associated proteins such as MAP103.

(14) Fig. 5F, G, S5F. Along the same lines, it would be helpful to show example images for each category - "kinetochores", "kinetochores + spindle", and "spindle".

As suggested by the reviewer, we have now included example images for each category (‘kinetochores’, ‘kinetochores + spindle’, ‘spindle’) along with a schematic illustration in Fig. 5F.

(15) Line 332 and Fig. S6A. The experiment may be repeated in the presence of ATP or nonhydrolyzable ATP analogs.

We thank the reviewer for the suggestion. We envisage such experiments for an in-depth follow-up study.

(16) Line 342, "motor activity of KIN-A". Until KIN-A is shown to have motor activity, the result based on the rigor mutant does not show that the motor activity of KIN-A promotes chromosome congression. The result suggests that the ATPase activity of KIN-A is important.

We changed that sentence as suggested by the reviewer.

(17) Line 419 -. The authors base their discussion on the speculation that KIN-A is a plus-end directed motor. Please justify this speculation.

Indeed, the notion that KIN-A is a plus-end directed motor remains a hypothesis, which is based on sequence alignments with other plus-end directed motors and the observation that the KIN-A motor domain is involved in translocation of the CPC to the central spindle in anaphase. We have modified the corresponding section in the discussion as follows:

‘It remains to be investigated whether KIN-A truly functions as a plus-end directed motor. The role of the KIN-B in this context is equally unclear. Since KIN-B does not possess a functional kinesin motor domain, we deem it unlikely that the KIN-A:KIN-B heterodimer moves hand-over-hand along microtubules as do conventional (kinesin-1 family) kinesins. Rather, the KIN-A motor domain may function as a single-headed unit and drive processive plus-end directed motion using a mechanism similar to the kinesin-3 family kinesin KIF1A (Okada and Hirokawa, 1999).’

(18) Line 422-423, "plus-end directed motion using a mechanism similar to kinesin-3 family kinesins (such as KIF1A)." Please cite a reference supporting this statement.

See above. We cited a paper by (Okada and Hirokawa, 1999).

**Reviewer #2 (Recommendations For The Authors):**
Please provide a quantification of data shown in Figure 2F-H and described in lines 151-166.

We decided not to provide a quantification for these data due to low sample sizes for some of the constructs (e.g. expression not observed in all cells).

It appears as if the paper more or less follows a chronological order of the experiments that were performed before AF multimer enabled the insightful and compelling structural analysis. That is a matter of style, but in some cases, the writing could be updated, shortened, or re-arranged into a more logical order. Concrete examples:(i) Line 144: "we did not include CPC2 for further analysis in this study"Although CPC2 features at a prominent and interesting position in the predicted structures of the kinetoplastid CPC, shown in later main figures.

We attempted RNAi-mediated depletion of CPC2 using two different shRNA constructs. However, we cannot exclude the possibility that the knockdown of CPC2 was less efficient compared with the other CPC subunits. For this reason, we decided to remove all the data on CPC2 from Fig. S2.

(ii) The work with the KIN-A motor domain only and KIN-A ∆motor domain (Fig 2) begs the question about a more subtle mutation to interfere with the motor domain. Which is ultimately presented in Fig 6. I think that the final paragraph and Figure 6 follow naturally after Figure 2.

We appreciate the suggestion. However, we would like to keep Figure 6 there.

(iii) The high-confidence structural predictions in Fig 3 and Fig 4 are insightful. The XL-MS descriptions that precede them are not so helpful (Fig 3A and 4G and in the text). To emphasize their status as experimental support for the predicted structures, which is very important, it would be good to discuss the XL-MS after presenting the models.

As suggested, we have re-arranged the text and/or figures such that the AF2 predictions are discussed first and the CLMS data are brought in afterwards.

Figure 1A prominently features an arbitrary color code and a lot of protein IDs without a legend. That is not a very convincing start. Figure S1 is more informative, containing annotated protein names and results of the KIN-A and KIN-B IPs. Please improve Figure 1A, for example by presenting a modified version of Figure S1. In all these types of figures, please list both protein names and gene IDs.

We agree with the reviewer that the IP-MS data in Fig. S1 is more informative and hence decided to swap the heatmaps in Fig. 1A and Fig. S1A. We further annotated the heatmap corresponding to the Aurora BAUK1 IP-MS (now presented in Fig. S1) as suggested by the reviewer.

The visualization of the structural predictions is not consistent among figures:(i) The structure in Fig 4I is important and could be displayed larger. The pLDDT scores, and especially those of the non-displayed models, do not add much information and should not be a main panel. If the authors want to display the pLDDT scores, I recommend a panel (main or supplement) of the structure colored for local prediction confidences, as in Fig 5A.(ii) In Figure 5A itself, it is hard to follow the chains in general, and KIN-A in particular, since the structure is pLDDT-coloured. Please present an additional panel colored by chain (consistent with Fig 4I, as mentioned above).(iii) The summarizing diagram, currently displayed as Fig 4J, should be placed after Fig 5A and take the discovered KIN-A - KKT9-11 connection into account. Ideally, it also covers the suspected importance of the motor domain and serves as a summarising diagram.

We thank the reviewer for the constructive comments. For each structure prediction, we now present two images side by side; one coloured by chain and one colored by pLDDT. We recently re-ran AF2 for the full CPC and also for the KKT7N-KKT8 complex, and got improved predictions. Hence some of the models in Fig. 3/S3 and Fig. 4/S4 have been updated accordingly. For the CLMS plots, we also decided to colour the cross-links according to whether the 30 angstrom distance constraints were fulfilled or not in the AF2 prediction. We also increased the size of the structures shown in Fig. 4. Furthermore, we decided to remove the summarizing diagram from Fig. 4 and instead made a new main Fig. 7, which shows a more detailed schematic, which also takes into account the proposed function of the KIN-A motor domain, as suggested by the reviewer, and other points addressed in the Discussion.

The methods section for the structural predictions lacks essential information. Predictions can only be reproduced if the version of AF2 multimer v2.x is specified and key parameters are mentioned.

As suggested, we have added the details in the Materials and Methods section as follows.

‘Structural predictions of KIN-A/KIN-B, KIN-A310-862/KIN-B317-624, CPC1/CPC2/KIN-A300-599/KIN-B 317-624, and KIN-A700-800/KKT9/KKT11 were performed using ColabFold version 1.3.0 (AlphaFold-Multimer version 2), while those of AUK1/CPC1/CPC2/KIN-A1-599/KIN-B, KKT71-261/KKT9/KKT11/KKT8/KKT12, KKT9/KKT11/KKT8/KKT12, and KKT71-261/KKT9/KKT11 were performed using ColabFold version 1.5.3 (AlphaFold-Multimer version 2.3.1) using default settings, accessed via https://colab.research.google.com/github/sokrypton/ColabFold/blob/v1.3.0/AlphaFold2.ipynb and https://colab.research.google.com/github/sokrypton/ColabFold/blob/v1.5.3/AlphaFold2.ipynb.’

Line 121, please explain the "Unexpectedly" by including a reference to the work from Li and colleagues. A statement with some details would be useful, as the difference between both studies appears to be crucial for the novelty of this paper. Alternatively, refer to this being covered in the discussion.

To clarify this point, we modified this paragraph in the results section:

‘To our surprise, KIN-A-YFP and GFP-KIN-B exhibited a CPC-like localization pattern identical to that of Aurora BAUK1: Both kinesins localized to kinetochores from S phase to metaphase, and then translocated to the central spindle in anaphase (Figs. 1, C-E). Moreover, like Aurora BAUK1, a population of KIN-A and KIN-B localized at the new FAZ tip from late anaphase onwards (Figs. S1, B and C). This was unexpected, because KIN-A and KIN-B were previously reported to localize to the spindle but not to kinetochores or the new FAZ tip (Li et al., 2008). These data suggest that KIN-A and KIN-B are bona fide CPC proteins in trypanosomes, associating with AuroraAUK1, INCENPCPC1 and CPC2 throughout the cell cycle.’

Line 285 refers to "conserved" regions in the C-terminal part of KIN-A, referring to Figure 5. Please expand the MSA in Figure 5B to get an idea about the conservation/variation outside CD1 and CD2.

We now present the full MSA for KIN-A proteins in kinetoplastids in Fig. S5A.

Please specify what is meant by Line 367-369 for someone who is not familiar with the work by Komaki et al. 2022. Either clarify in the text or clarify in the text with data to support it.

We updated the corresponding section in the discussion as follows:

‘Komaki et al. recently identified two functionally redundant CPC proteins in Arabidopsis, Borealin Related Interactor 1 and 2 (BORI1 and 2), which engage in a triple helix bundle with INCENP and Borealin using a conserved helical domain but employ an FHA domain instead of a BIR domain to read H3T3ph (Komaki et al., 2022).’

Data presented in Figure S6A, the microtubule co-sedimentation assay, is not convincing since a substantial amount of KIN-A/B is pelleted in the absence of microtubules. Did the authors spin the proteins in BRB80 before the assay to continue with soluble material and reduce sedimentation in the absence of microtubules? If the authors want to keep the wording in lines 331-332, the MT-binding properties of KIN-A and KIN-B need to be investigated in more detail, for example with a titration and a quantification thereof. Otherwise, they should change the text and replace "confirms" with "is consistent with". In any case, the legend needs to be expanded to include more information.

To address the point above, we have added the following text in the legend corresponding to Fig. S6:

‘Microtubule co-sedimentation assay with 6HIS-KIN-A2-309 (left) and 6HIS-KIN-B2-316 (right). S and P correspond to supernatant and pellet fractions, respectively. Note that both constructs to some extent sedimented even in the absence of microtubules. Hence, lack of microtubule binding for KIN-B may be due to the unstable non-functional protein used in this study.’

We have also updated the main text in the results section:

‘We therefore speculated that anaphase translocation of the kinetoplastid CPC to the central spindle may involve the kinesin motor domain of KIN-A. KIN-B is unlikely to be a functional kinesin based on the absence of several well-conserved residues and motifs within the motor domain, which are fully present in KIN-A (Li et al., 2008). These include the P-loop, switch I and switch II motifs, which form the nucleotide binding cleft, and many conserved residues within the α4-L12 elements, which interact with tubulin (Fig. S6A) (Endow et al., 2010). Consistent with this, the motor domain of KIN-B, contrary to KIN-A, failed to localize to the mitotic spindle when expressed ectopically (Fig. S2E) and did not co-sediment with microtubules in our in vitro assay (Fig. S6B).’

Details:The readability of the pAE plots could be improved by arranging sequences according to their position in the structure. For example in Fig4I, KKT8 could precede KKT12. If it is easy to update this, the authors might want to do so.

We re-ran the AF2 predictions for the KKT7N – KKT8 complex in Fig. 4/S4 and changed the order according to the reviewer’s suggestion (KKT9:KKT11:KKT8:KKT12).

The same paper is referred to as Je Van Hooff et al. 2017 and as Van Hooff et al. 2017

Thank you for pointing this out. We have corrected the citation.

**Reviewer #3 (Recommendations For The Authors):**
(1) Please state at the end of the introduction/start of the results section that this work was performed in procyclic trypanosomes. Given that the cell cycles of procyclic and bloodstream forms differ, this is important.

We added this information at the end of the introduction:

‘Here, by combining biochemical, structural and cell biological approaches in procyclic form *T. brucei*, we show that the trypanosome CPC is a pentameric complex comprising Aurora BAUK1, INCENPCPC1, CPC2 and the two orphan kinesins KIN-A and KIN-B.’

(2) Please define NLS at first use (line 118), and for clarity, explain the rationale for using GFP with an NLS.

NLS refers to a short ‘nuclear localization signal’ (TGRGHKRSREQ) (Marchetti et al., 2000), which ensures that the ectopically expressed construct is imported into the nucleus. When we previously expressed truncations of KKT2 and KKT3 kinetochore proteins, many fragments did not go into the nucleus presumably due to the lack of an NLS, which prevented us from determining which domains are responsible for their kinetochore localization. We have since then consistently used this short NLS sequence in our inducible GFP fusions in the past without any complications. We added a sentence in the Materials & Methods section under Trypanosome culture: ‘All constructs for ectopic expression of GFP fusion proteins include a short nuclear localization signal (NLS) (Marchetti et al., 2000).’ To avoid unnecessary confusion, we removed ‘NLS’ from the main text and figures.

(3) Lines 148-150 - it would strengthen this claim if KIN-A/B protein levels were assessed by Western blot.

We now present a Western blot in Fig. S2C, showing that bulk KIN-B levels are clearly reduced upon KIN-A RNAi. The same is true also to some extent for KIN-A levels upon KIN-B RNAi, although this is less obvious, possibly due to the lower efficiency of KIN-B compared to KIN-A RNAi as judged by fluorescence microscopy (quantified in Fig. 2D and 2E).

(4) Line 253 - the text mentions the removal of both KKT9 and KKT11, which is not consistent with the figure (Fig 4H) - do you mean the removal of either KKT9 or KKT11?

Yes, we thank the reviewer for pointing out this mistake in the text, which has now been corrected.

(5) Line 337 - please include a reference for the G209A ATPase-defective rigor mutant - has this been shown to result in KIN-A being inactive previously?

Please see above our answer in public review.

(6) It is not always obvious when fluorescent fusion proteins are being expressed endogenously or ectopically, or when they are being expressed in an RNAi background or not without tracing the cell lines in Table S1 - please ensure this is clearly stated throughout the manuscript.

We now made sure that this is clearly stated in the main text as well as in the figure legends.

(7) Line 410 - 'KIN-A C-terminal tail is stuffed full of conserved CDK1CRK3 sites' - what does 'stuffed full' really mean (this is rather imprecise) and what are the consensus sites - are these CDK1 consensus sites that are assumed to be conserved for CRK3? I'm not aware of consensus sites for CRK3 having been determined, but if they have, this should be referenced.

We have modified the corresponding section in the discussion as follows:

‘In support of this, the KIN-A C-terminal tail harbours many putative CRK3 sites (10 sites matching the minimal S/T-P consensus motif for CDKs) and is also heavily phosphorylated by Aurora BAUK1 in vitro (Ballmer et al. 2024). Finally, we speculate that the interaction of KIN-A motor domain with microtubules, coupled to the force generating ATP hydrolysis and possibly plus-end directed motion, eventually outcompetes the weakened interactions of the CPC with the kinetochore and facilitates the extraction of the CPC from chromosomes onto spindle microtubules during anaphase. Indeed, deletion of the KIN-A motor domain or impairment of its motor function through N-terminal GFP tagging causes the CPC to be trapped at kinetochores in anaphase. Central spindle localization is additionally dependent on the ATPase activity of the KIN-A motor domain as illustrated by the KIN-A rigor mutant.’

(8) Lines 412-416: this proposal is written rather definitively - given no motor activity has been demonstrated for KIN-A, please make clear that this is still just a theory.

See above.

(9) Fig 1: KKT2 is not highlighted in Fig 1A - given this has been used for colocalization in Fig 1C-E, was it recovered, and if not, why not? Fig 1B-E: the S phase/1K*1N terminology is somewhat misleading. Not all S phase cells will have elongated kinetoplasts - usually an asterisk is used to signify replicated DNA, not kinetoplast shape. If it is to be used here for elongation, then for consistency, N* should be used for G2/mitotic cells.

Fig. 1A (now Fig. S1A) only shows the tip 30 hits. KKT2 was indeed recovered with Aurora BAUK1 (see Table S2) and is often used as a kinetochore marker in trypanosomes by our lab and others since the signal of fluorescently tagged KKT2 is relatively bright and KKT2 localizes to centromeres throughout the cell cycle.

(10) A general comment for all image figures is that these do not have accompanying brightfield images and it is therefore difficult to know where the cell body is, or sometimes which nuclei and kinetoplasts belong to which cell where DNA from more than one cell is within the image. It would be beneficial if brightfield images could be added, or alternatively, the cell outlines were traced onto DAPI or merged images. Also, brightfield images would allow the stage of cytokinesis (pre-furrowing/furrowing/abscission) in anaphase cells to be determined.

Since this study primarily addresses the recruitment mechanism of the CPC to kinetochores and to the central spindle from S phase to metaphase and in anaphase, respectively, and CPC proteins are not observed outside of the nucleus during these cell cycle stages, we did not present brightfield images in the figures. However, this point is particularly valid for discerning the localization of KIN-A and KIN-B to the new FAZ tip from late anaphase onwards. Hence, we acquired new microscopy data for Fig. S1B and S1C, which now includes phase contrast images, and have chosen representative cells in late anaphase and telophase. We hope that the signal of Aurora BAUK1, KIN-A and KIN-B at the anterior end of the new FAZ can be now distinguished more clearly.

(11) Fig 2A: legend should state that the micrographs show the localisation of the proteins within the nucleus as whole cells are not shown. 2C: can INCENP not be split into 2 lines - the 'IN' looks like 1N at first glance, which is confusing.

We have applied the suggested change in Fig. 2.

(12) Fig 3 (and other AF2 figures): Could the lines for satisfied & not satisfied in the key be thicker so they more closely resemble the lines in the figure and are less likely to be confused with the disordered regions of the CPC components?

We have now made those lines thicker.

(13) Why were different E value thresholds used in Fig 3 and Fig 4?

The CLMS data in Fig. 3 and Fig. 4 now both use the same E value threshold of E-3 (previously E-4 was used in Fig. 4). To determine a sensible significance threshold, we included some yeast protein sequences (‘false positives’) in the database used in pLink2 for identification of crosslinked peptides.Note that we recently also re-ran AF2 for the full CPC and for the KKT7N-KKT8 complex and got improved predictions. Hence some of the models in Fig. 3/S3 and Fig. 4/S4 have been updated accordingly. For the CLMS plots, we also decided to colour the cross-links according to whether the 30 angstrom distance constraints were fulfilled or not in the AF2 prediction.

(14) Fig 4H legend - please give the expected sizes of these recombinant proteins & check the 3rd elution panel (see public review comments).

See above response in public review.

(15) Fig 4I - please explain what the colours of the PAE plot and the values in the key signify, as well as how the Scored Residue values are arrived at. Please also define the pIDDT in the legend.

We have cited DeepMind’s 2021 methods paper, in which the outputs of AlphaFold are explained in detail. We also added a short description of the pLDDT and PAE scores and the corresponding colour coding in the legends of Fig. 3 and Fig. 4, respectively.

From figure 3 legend:

‘(B) Cartoon representation showing two orientations of the trypanosome CPC, coloured by protein on the left (Aurora BAUK1: crimson, INCENPCPC1: green, CPC2: cyan, KIN-A: magenta, and KIN-B: yellow) or according to their pLDDT values on the right, assembled from AlphaFold2 predictions shown in Figure S3. The pLDDT score is a per-residue estimate of the confidence in the AlphaFold prediction on a scale from 0 – 100. pLDDT > 70 (blue, cyan) indicates a reasonable accuracy of the model, while pLDDT < 50 (red) indicates a low accuracy and often reflects disordered regions of the protein (Jumper et al., 2021). BS3 crosslinks in (B) were mapped onto the model using PyXlinkViewer (blue = distance constraints satisfied, red = distance constraints violated, Cα-Cα Euclidean distance threshold = 30 Å) (Schiffrin et al., 2020).’

From Figure 4 legend:

‘(G) AlphaFold2 model of the KKT7 – KKT8 complex, coloured by protein (KKT71-261: green, KKT8: blue, KKT12: pink, KKT9: cyan and KKT11: orange) (left) and by pLDDT (center). BS3 crosslinks in (H) were mapped onto the model using PyXlinkViewer (Schiffrin et al., 2020) (blue = distance constraints satisfied, red = distance constraints violated, Cα-Cα Euclidean distance threshold = 30 Å). Right: Predicted Aligned Error (PAE) plot of model shown on the left (rank_2). The colour indicates AlphaFold’s expected position error (blue = low, red = high) at the residue on the x axis if the predicted and true structures were aligned on the residue on the y axis (Jumper et al., 2021).’

(16) Fig 6 legend - Line 730 should say (F) not (C).

Thank you for pointing out this typo.

(17) Fig S1A - a key is missing for the colours. Fig S1B/C - cell outlines or a brightfield image are really needed here - see earlier comment. Fig S1D - there doesn't seem to be a method for how this tree was generated.

See above response in public review regarding Fig. S1A and S1B/C. The tree in Fig. S1D is based on (Butenko et al., 2020).

(18) Fig S2: A: how was protein knockdown validated (especially for CPC2 where there was little obvious phenotype)? Fig S2B: the y-axis should read proportion of cells, not percentage. Fig S2E - NLS should be labelled.

Thank you for pointing out the mistake in the labelling.

(19) Fig S3: PAE plots should be labelled with protein names, not A-E. Similarly, the pIDDT plots should be labelled as in Fig 4I.

We have corrected the labelling in Fig. S3.

(20) Fig S5A-D - cell cycle stage labels are missing from images.

Thank you for pointing out the missing cell cycle stage labels.

Addition by editor:In line 126 the statement that KIN-A and KIN-B "associate with Aurora-AUK1, INCENP-CPC1 and CPC2 throughout the cell cycle" seems too strong. There is no direct evidence for this. Please re-phrase as "likely associate" or "suggest... that ... may...".

We have modified that sentence according to the editor’s suggestion.

References:

Akiyoshi, B., and K. Gull. 2014. Discovery of Unconventional Kinetochores in Kinetoplastids. Cell. 156. doi:10.1016/j.cell.2014.01.049.

Butenko, A., F.R. Opperdoes, O. Flegontova, A. Horák, V. Hampl, P. Keeling, R.M.R. Gawryluk, D. Tikhonenkov, P. Flegontov, and J. Lukeš. 2020. Evolution of metabolic capabilities and molecular features of diplonemids, kinetoplastids, and euglenids. BMC Biology 2020 18:1. 18:1–28. doi:10.1186/S12915-020-0754-1.

Cormier, A., D.G. Drubin, and G. Barnes. 2013. Phosphorylation regulates kinase and microtubule binding activities of the budding yeast chromosomal passenger complex in vitro. J Biol Chem. 288:23203–23211. doi:10.1074/JBC.M113.491480.Endow, S.A., F.J. Kull, and H. Liu. 2010. Kinesins at a glance. J Cell Sci. 123:3420. doi:10.1242/JCS.064113.

Fink, S., K. Turnbull, A. Desai, and C.S. Campbell. 2017. An engineered minimal chromosomal passenger complex reveals a role for INCENP/Sli15 spindle association in chromosome biorientation. J Cell Biol. 216:911–923. doi:10.1083/JCB.201609123.

van der Horst, A., M.J.M. Vromans, K. Bouwman, M.S. van der Waal, M.A. Hadders, and S.M.A. Lens. 2015. Inter-domain Cooperation in INCENP Promotes Aurora B Relocation from Centromeres to Microtubules. Cell Rep. 12:380–387. doi:10.1016/J.CELREP.2015.06.038.

Ishii, M., and B. Akiyoshi. 2020. Characterization of unconventional kinetochore kinases KKT10/19 in *Trypanosoma brucei*. J Cell Sci. doi:10.1242/jcs.240978.

Jeyaprakash, A.A., C. Basquin, U. Jayachandran, and E. Conti. 2011. Structural Basis for the Recognition of Phosphorylated Histone H3 by the Survivin Subunit of the Chromosomal Passenger Complex. Structure. 19:1625–1634. doi:10.1016/J.STR.2011.09.002.

Jeyaprakash, A.A., U.R. Klein, D. Lindner, J. Ebert, E.A. Nigg, and E. Conti. 2007. Structure of a Survivin–Borealin–INCENP Core Complex Reveals How Chromosomal Passengers Travel Together. Cell. 131. doi:10.1016/j.cell.2007.07.045.

Jumper, J., R. Evans, A. Pritzel, T. Green, M. Figurnov, O. Ronneberger, K. Tunyasuvunakool, R. Bates, A. Žídek, A. Potapenko, A. Bridgland, C. Meyer, S.A.A. Kohl, A.J. Ballard, A. Cowie, B. Romera-Paredes, S. Nikolov, R. Jain, J. Adler, T. Back, S. Petersen, D. Reiman, E. Clancy, M. Zielinski, M. Steinegger, M. Pacholska, T. Berghammer, S. Bodenstein, D. Silver, O. Vinyals, A.W. Senior, K. Kavukcuoglu, P. Kohli, and D. Hassabis. 2021. Highly accurate protein structure prediction with AlphaFold. Nature 2021 596:7873. 596:583–589. doi:10.1038/s41586-021-03819-2.

Kang, J.S., I.M. Cheeseman, G. Kallstrom, S. Velmurugan, G. Barnes, and C.S.M. Chan. 2001. Functional cooperation of Dam1, Ipl1, and the inner centromere protein (INCENP)-related protein Sli15 during chromosome segregation. J Cell Biol. 155:763–774. doi:10.1083/JCB.200105029.

Klein, U.R., E.A. Nigg, and U. Gruneberg. 2006. Centromere targeting of the chromosomal passenger complex requires a ternary subcomplex of Borealin, Survivin, and the N-terminal domain of INCENP. Mol Biol Cell. 17:2547–2558. doi:10.1091/MBC.E05-12-1133.

Komaki, S., E.C. Tromer, G. De Jaeger, N. De Winne, M. Heese, and A. Schnittger. 2022. Molecular convergence by differential domain acquisition is a hallmark of chromosomal passenger complex evolution. Proc Natl Acad Sci U S A. 119. doi:10.1073/PNAS.2200108119/-/DCSUPPLEMENTAL.

Li, Z. 2012. Regulation of the Cell Division Cycle in *Trypanosoma brucei*. Eukaryot Cell. 11:1180. doi:10.1128/EC.00145-12.

Li, Z., J.H. Lee, F. Chu, A.L. Burlingame, A. Günzl, and C.C. Wang. 2008. Identification of a Novel Chromosomal Passenger Complex and Its Unique Localization during Cytokinesis in *Trypanosoma brucei*. PLoS One. 3. doi:10.1371/journal.pone.0002354.

Mackay, A.M., D.M. Eckley, C. Chue, and W.C. Earnshaw. 1993. Molecular analysis of the INCENPs (inner centromere proteins): separate domains are required for association with microtubules during interphase and with the central spindle during anaphase. J Cell Biol. 123:373–385. doi:10.1083/JCB.123.2.373.

Marchetti, M.A., C. Tschudi, H. Kwon, S.L. Wolin, and E. Ullu. 2000. Import of proteins into the trypanosome nucleus and their distribution at karyokinesis. J Cell Sci. 113 ( Pt 5):899–906. doi:10.1242/JCS.113.5.899.

Nakajima, Y., A. Cormier, R.G. Tyers, A. Pigula, Y. Peng, D.G. Drubin, and G. Barnes. 2011. Ipl1/Aurora-dependent phosphorylation of Sli15/INCENP regulates CPC-spindle interaction to ensure proper microtubule dynamics. J Cell Biol. 194:137–153. doi:10.1083/JCB.201009137.

Noujaim, M., S. Bechstedt, M. Wieczorek, and G.J. Brouhard. 2014. Microtubules accelerate the kinase activity of Aurora-B by a reduction in dimensionality. PLoS One. 9. doi:10.1371/JOURNAL.PONE.0086786.

Okada, Y., and N. Hirokawa. 1999. A processive single-headed motor: Kinesin superfamily protein KIF1A. Science (1979). 283:1152–1157. doi:10.1126/SCIENCE.283.5405.1152.

Rice, S., A.W. Lin, D. Safer, C.L. Hart, N. Naber, B.O. Carragher, S.M. Cain, E. Pechatnikova, E.M. Wilson-Kubalek, M. Whittaker, E. Pate, R. Cooke, E.W. Taylor, R.A. Milligan, and R.D. Vale. 1999. A structural change in the kinesin motor protein that drives motility. Nature 1999 402:6763. 402:778–784. doi:10.1038/45483.

Sablin, E.P., F.J. Kull, R. Cooke, R.D. Vale, and R.J. Fletterick. 1996. Crystal structure of the motor domain of the kinesin-related motor ncd. Nature 1996 380:6574. 380:555–559. doi:10.1038/380555a0.

Samejima, K., M. Platani, M. Wolny, H. Ogawa, G. Vargiu, P.J. Knight, M. Peckham, and W.C. Earnshaw. 2015. The Inner Centromere Protein (INCENP) Coil Is a Single α-Helix (SAH) Domain That Binds Directly to Microtubules and Is Important for Chromosome Passenger Complex (CPC) Localization and Function in Mitosis. J Biol Chem. 290:21460–21472. doi:10.1074/JBC.M115.645317.

Schiffrin, B., S.E. Radford, D.J. Brockwell, and A.N. Calabrese. 2020. PyXlinkViewer: A flexible tool for visualization of protein chemical crosslinking data within the PyMOL molecular graphics system. Protein Sci. 29:1851–1857. doi:10.1002/PRO.3902.